# Direct photoresponsive inhibition of a p53-like transcription activation domain in PIF3 by *Arabidopsis* phytochrome B

Chan Yul Yoo [1,6], Jiangman He [1,6], Qing Sang[1,2,6], Yongjian Qiu [1], Lingyun Long[1], Ruth Jean-Ae Kim[1], Emily G. Chong[1], Joseph Hahm[1], Nicholas Morffy[3], Pei Zhou [4], Lucia C. Strader[3], Akira Nagatani[5], Beixin Mo [2], Xuemei Chen [1] & Meng Chen [1✉]

Photoactivated phytochrome B (PHYB) binds to antagonistically acting PHYTOCHROME-INTERACTING transcription FACTORs (PIFs) to regulate hundreds of light responsive genes in *Arabidopsis* by promoting PIF degradation. However, whether PHYB directly controls the transactivation activity of PIFs remains ambiguous. Here we show that the prototypic PIF, PIF3, possesses a p53-like transcription activation domain (AD) consisting of a hydrophobic activator motif flanked by acidic residues. A PIF3mAD mutant, in which the activator motif is replaced with alanines, fails to activate PIF3 target genes in *Arabidopsis*, validating the functions of the PIF3 AD in vivo. Intriguingly, the N-terminal photosensory module of PHYB binds immediately adjacent to the PIF3 AD to repress PIF3's transactivation activity, demonstrating a novel PHYB signaling mechanism through direct interference of the transactivation activity of PIF3. Our findings indicate that PHYB, likely also PHYA, controls the stability and activity of PIFs via structurally separable dual signaling mechanisms.

[1] Department of Botany and Plant Sciences, Institute for Integrative Genome Biology, University of California, Riverside, CA, USA. [2] Guangdong Provincial Key Laboratory for Plant Epigenetics, College of Life Sciences and Oceanography, Shenzhen University, Shenzhen, China. [3] Department of Biology, Duke University, Durham, NC, USA. [4] Department of Biochemistry, Duke University School of Medicine, Durham, NC, USA. [5] Department of Botany, Graduate School of Science, Kyoto University, Sakyo-ku, Kyoto, Japan. [6] These authors contributed equally: Chan Yul Yoo, Jiangman He, Qing Sang. ✉email: meng.chen@ucr.edu

The plant genome is exquisitely sensitive to and can be transcriptionally reprogrammed by changes in light intensity and composition[1–3]. In particular, alterations in the intensity and ratio of red (R, 600–700 nm) and far-red (FR, 700–750 nm) light are powerful environmental cues that inform about space and time—such as the availability of photosynthetically active R-light radiation, the sense of morning and evening as well as seasonal fluctuations in day length, and the threat by neighboring vegetation or shade that depletes R light and alters the R-to-FR ratio[4–9]. Plants detect R and FR light through evolutionarily conserved photoreceptors called phytochromes (PHYs)[10]. The prototypical plant PHY is a homodimer: each monomer comprises an N-terminal photosensory module and a C-terminal signaling output module[11,12]. The N-terminal photosensory module consists of four domains: an N-terminal extension, a PAS (Period/ARNT/Single-minded) domain, a GAF (cyclic GMP-regulated cyclic nucleotide phosphodiesterase/ Adenylate cyclase/FhlA) domain that binds a tetrapyrrole chromophore, and a PHY (phytochrome-specific) domain. At the PAS-GAF interface, there is an unusual figure-eight knot called the "light-sensing knot", which is tightly linked to the chromophore and plays an important role in signaling[13–15]. The C-terminal output module contains two tandem PAS domains and a histidine kinase-related domain (HKRD), which collectively mediate dimerization, subcellular localization, and signaling[16–19]. Absorption of R or FR light by the tetrapyrrole moiety embedded in the N-terminal photosensory module photoconverts PHYs between two relatively stable forms: the R-light-absorbing inactive Pr and the FR-light-absorbing active Pfr[11,12]. The conformational changes of PHYs dictate their subcellular localization: while the Pr localizes in the cytoplasm, the Pfr accumulates in the nucleus and forms PHY-containing subnuclear foci called photobodies, concomitantly reprogramming the expression of hundreds of light-responsive genes[20,21]. As such, the equilibrium of the Pr and Pfr of PHYs quantitatively connects environmental light cues to light-responsive gene expression. PHY-mediated gene regulation is at the center of plant-environment interactions and profoundly impacts all aspects of plant development, growth, metabolism, and immunity, enabling plants to thrive in complex natural environments[22].

In the plant reference species *Arabidopsis thaliana* (*Arabidopsis*), PHYs are encoded by a small gene family, *PHYA-E*[23]. PHYB is the most prominent PHY, evidently because only PHYB directly interact with an entire family of basic/helix-loop-helix (bHLH) transcription factors named PHYTOCHROME-INTERACTING FACTORs (PIFs)[24]. The PIF family comprises eight members: PIF1-8 (PIF2 and PIF6 are also called PIL1 and PIL2 [PIF3-Like1 and 2], respectively)[25,26]. The C-termini of all PIFs contain a bHLH domain for dimerization and DNA binding to the G-box as well as the PIF-binding E-box (PBE-box)[27,28], while the N-termini contain an Active-PHYB Binding (APB) motif, which interacts preferentially with the light-sensing knot in the N-terminal photosensory module of photoactivated PHYB[15,24]. PIF1 and PIF3 also interact with the Pfr form of PHYA via a separate Active-PHYA Binding (APA) motif, which is absent in the rest of the PIFs[26]. In general, PIFs play antagonistic roles in PHYB signaling[29,30]. During seedling development, after seed germination underground in the absence of light, four PIFs, PIF1, PIF3, PIF4, and PIF5, promote the dark-grown developmental program called skotomorphogenesis by blocking leaf development and chloroplast biogenesis as well as accelerating the elongation of the embryonic stem (hypocotyl)—a set of coordinated actions allowing seedlings to quickly emerge from the soil[31,32]. To promote hypocotyl elongation, PIFs bind to the enhancer regions of growth-relevant genes, such as those involved in the biosynthesis and signaling of the plant growth hormone

auxin, and activate their transcription[27,28]. Upon seedling emergence from the ground and exposure to light, photoactivated PHYB translocates to the nucleus and interacts with PIF1, PIF3, PIF4, and PIF5 to promote their rapid phosphorylation, ubiquitylation, and degradation by the proteasome, thereby reprogramming the expression of light-responsive PIF target genes[19,33–35]. PHYB-triggered degradation of PIFs has been considered a central mechanism of PHYB signaling[26], implying that PHYB regulates PIF target genes mainly indirectly by controlling the abundance of PIFs.

Accumulating circumstantial evidence suggests that PHYB may also directly control the transcriptional activity of PIFs. First, not all PIFs are degraded in the light or by photoactivated PHYB. For example, the protein level of PIF7 is not significantly different between dark and light conditions[36,37]. Also, although PIF4 and PIF5 rapidly diminish during the initial exposure to light, when seedlings are grown under diurnal or continuous light conditions, both can accumulate to high levels during the daytime, when PHYB is active[38,39]. Moreover, the stability of PIFs varies at different temperatures. For example, PIF3, despite being almost undetectable in ambient temperatures, becomes stabilized in the light in cold temperatures[40]. Therefore, the PHYB-PIF interaction could regulate the activity of PIFs in light conditions. Second, although PHYB triggers the light-induced degradation of PIF3, recent studies surprisingly revealed that the light-dependent interaction between the light-sensing knot of PHYB and the APB of PIF3 is not responsible for PIF3 degradation[16,41,42]. Instead, PIF3 degradation relies on the weaker, light-independent interaction with PHYB's C-terminal output module[16,41,42]. These results leave open the important question of what the functional significance of the PHYB-APB interaction is in PIF3 regulation. One proposed hypothesis is that binding of PHYB to the APB regulates the transcriptional activity of PIFs[41,43]. Supporting this hypothesis, the PHYB-APB interaction was shown to attenuate PIF3's DNA-binding activity independently of its degradation[41,43]. However, because the APB motif is located at the very N-terminus of PIF3, far from the C-terminal bHLH domain, and PHYB does not interact directly with the bHLH[24,27], it remains unclear whether PHYB influences the DNA binding of PIF3 directly or indirectly via the control of its transactivation activity.

The main roadblock that hinders our understanding of the relationship between PHYB and the transactivation activity of PIFs is the lack of a molecular understanding of PIFs' transcription activation domains (ADs). It is well understood that transcriptional activators follow a modular design consisting of a DNA-binding domain to selectively bind to specific DNA sequences (enhancers) and an AD to initiate transcription[44]. Although the classes of DNA-binding domains have been well defined and can be recognized based on conserved amino acid sequences[45], ADs remain difficult to predict due to their poor sequence conservation[46–48]. As a result, despite PIFs being widely accepted as transcriptional activators[28], their AD sequences have not been precisely defined. A previous attempt to map the AD of PIF3 identified two separate regions in its N-terminal half that carry transactivation activity in yeast, although these putative ADs have not been validated in planta[49]. To further explore the possibility of a direct link between PHYB and the transactivation activity of PIFs, first, we utilized truncation analysis and alanine-scanning mutagenesis combined with yeast and in planta assays to determine the AD of PIF3. Our results show that PIF3 possesses a single AD, which surprisingly resembles the ADs of the mammalian tumor suppressor p53 and the yeast activator Gcn4, revealing the unexpected conservation of sequence-specific ADs across the animal, fungal, and plant kingdoms. We demonstrate that binding of PHYB's N-terminal

photosensory module to the APB motif of PIF3 represses PIF3's transactivation activity, revealing a new PHYB signaling mechanism via direct photoinhibition of the transactivation activity of PIFs.

## Results

**The aa$_{91-114}$ region of PIF3 confers AD activity in yeast**. To define and validate the AD of PIF3, ideally, it requires evaluation of the in planta functions of a PIF3 mutant specifically lacking its AD activity while leaving other biochemical functions intact. A previous study identified two PIF3 regions with transactivation activity in yeast using a modified yeast one-hybrid screen and random mutagenesis[49]. However, the PIF3 mutants isolated via this approach contained multiple mutations; therefore, the residues directly and specifically required for the AD activity could not be unambiguously pinpointed[49]. To circumvent these potential issues, we chose to systematically map the region and residues that are both necessary and sufficient for PIF3's transactivation activity by truncation analysis and targeted mutagenesis. Because PIF3 has been shown to be a potent activator in yeast[49], we first used yeast transactivation assays to examine PIF3's AD activity. We fused the Gal4 DNA-binding domain (DBD) to either the full-length PIF3 or a series of N- or C-terminal PIF3 truncation fragments in the bait construct of the Matchmaker Gold yeast two-hybrid system (Fig. 1a). We evaluated the self-activation activity of the recombinant DBD-PIF3 proteins using two assays: (1) a cell viability assay testing the activation of the Aureobasidin A (AbA) resistance gene *AUR1-C* and (2) a liquid assay quantifying the activation of a second reporter, β-galactosidase (Fig. 1a). Interestingly, all PIF3 fragments containing the region between amino acids 91 and 114 (aa$_{91-114}$)—i.e., PIF3-N1 to PIF3-N4, as well as PIF3-C1 and PIF3-C2—retained at least 87% of the activity of full-length PIF3 (Fig. 1a). In contrast, the two fragments missing part of aa$_{91-114}$— i.e., PIF3-N5 and PIF3-C3—completely lost the activity (Fig. 1a). The aa$_{91-114}$ fragment alone (PIF3-M2) was sufficient to activate gene expression, albeit with reduced activity (Fig. 1a). The weaker activity of aa$_{91-114}$ could be due to a lack of structural support provided by the adjacent protein context because a larger fragment containing amino acids 76–114 (PIF3-M1) displayed significantly higher activity (Fig. 1a).

Our results of aa$_{91-114}$ being an activating region is consistent with the previous study, which showed that the slightly larger aa$_{90-120}$ region carries transactivation activity[49]. However, we did not detect any transactivation activity between amino acids 27 and 43, which encompass the second activating region suggested by the previous study[49]. None of the four N-terminal fragments of PIF3 containing the second putative activating region, aa$_{1-52}$, aa$_{1-76}$, aa$_{1-90}$, and aa$_{1-101}$, showed any transactivation activity in our assays (Fig. 1a and Supplementary Fig. 1). To further resolve this discrepancy, we performed alanine-scanning mutagenesis in the region between amino acids 26 and 114 in the PIF3-N1 (aa$_{1-331}$) fragment by substituting every five consecutive amino acids with five alanines (Fig. 1b). Corroborating the results of the truncation analysis, only the alanine substitution mutants in the aa$_{91-114}$ region (m14–m18) completely abrogated the transactivation activity (Fig. 1b). In contrast, the mutants with mutations in the second suggested activating region (m1–m4) remained active, despite m1 and m2 showing reduced activities (Fig. 1b), indicating that the aa$_{26-43}$ region is not essential for PIF3's transactivation activity. Together, the results of the truncation analysis and alanine-scanning mutagenesis indicate that aa$_{91-114}$ is the only region both required and sufficient for PIF3's transactivation activity, and therefore, confers PIF3's AD function in yeast.

**PIF3 possesses a p53-like AD**. Studies of animal and yeast activators suggest that one class of ADs comprises a short, sequence-specific activator motif of hydrophobic residues embedded in an intrinsically disordered region enriched in acidic residues[47,48,50–52]. One identified activator motif is ΦxxΦΦ, where Φ indicates a bulky hydrophobic residue and x is any other amino acid[47]. For example, both ADs in the mammalian tumor suppressor p53, AD1 and AD2, contain the ΦxxΦΦ motif[50]. The same motif is also found in the yeast activator Gcn4[47,48]. Surprisingly, aligning the PIF3 AD to the ADs of p53 and Gcn4 revealed striking similarities: PIF3's AD features a **F**VP**WL** motif flanked by negatively charged aspartate and glutamate residues (Fig. 2a). The FVPWL motif in PIF3 shares the same hydrophobic residues as the activator motifs in p53 and Gcn4, including the two bulky aromatic amino acids F93 and W96. Both the activator motif and the negatively charged residues have been shown to play important roles in transactivation. For example, the ΦxxΦΦ motif in AD1 and AD2 of p53 mediates hydrophobic interactions with the transcriptional coactivator cyclic-AMP response element-binding protein (CREB)-binding protein (CBP) and its paralog p300[50]. The surrounding acidic residues participate in electrostatic interactions with transcriptional coactivators and create an intrinsically disordered context in the unbound state to enhance the accessibility of the activator motif[51]. Both the ΦxxΦΦ motif and surrounding acidic residues are highly conserved in PIF3 orthologs in eudicots (Fig. 2a), supporting the idea that PIF3 possesses a p53-like AD.

To evaluate the contribution of the conserved residues around the ΦxxΦΦ motif to PIF3's transactivation activity, we mutated amino acids 91 to 100 individually to alanine in the PIF3-N1 fragment. However, the single-amino-acid substitution mutants had little effect on PIF3's AD activity (Supplementary Fig. 2a, b), suggesting that the main functional and/or structural attributes of the AD are fulfilled redundantly by multiple residues. We then replaced the five residues of the FVPWL motif with alanines (a combination that was not included in the alanine-scanning mutagenesis) in full-length PIF3 and named it PIF3mAD (Fig. 2b). PIF3mAD completely lost its transactivation activity, confirming that this is the only AD in PIF3 (Fig. 2c). In addition, replacing only the three key hydrophobic residues in the ΦxxΦΦ motif, namely F93, W96, and L97, with arginine or serine also abolished AD activity (Supplementary Fig. 2a, c); similar mutations were shown to impede the activity of p53[53], suggesting that the critical roles of these hydrophobic residues in AD activity are conserved in PIF3.

The AD region of PIF3 was previously recognized as a motif of unknown function, which is moderately conserved among PIF family members (Fig. 2b)[54]. The ΦxxΦΦ motif can be recognized in PIF1, PIF4, PIF5, PIF7, and PIF8. While PIF1 and PIF8 contain a perfectly composed ΦxxΦΦ motif, PIF4, PIF5, and PIF7 are missing the initial hydrophobic residue (Fig. 2b). The flanking acidic residues are conserved to various degrees among PIF1, PIF4, PIF5, PIF7, and PIF8. In PIF1, PIF4, and PIF5, multiple acidic residues are present on both sides of the ΦxxΦΦ motif, whereas, in PIF7 and PIF8, at least one side of the ΦxxΦΦ motif contains significantly fewer acidic residues (Fig. 2b). The attributes of the PIF3 AD are least conserved in PIL1/PIF2 and PIL2/PIF6, in which the ΦxxΦΦ motif is barely recognizable and missing the potentially critical aromatic amino acids (Fig. 2b). We then examined the transactivation activities of all PIF members in yeast by fusing Gal4 DBD to the full-length PIF sequences. The PIF3 paralogs displayed a wide range of transactivation activities (Fig. 2c). The differences in activity could be, at least in part, explained by the degree of conservation of the two key attributes in the AD region. For example, PIF1, PIF4, PIF5, PIF7, and PIF8, which contain recognizable attributes of the PIF3 AD, exhibited

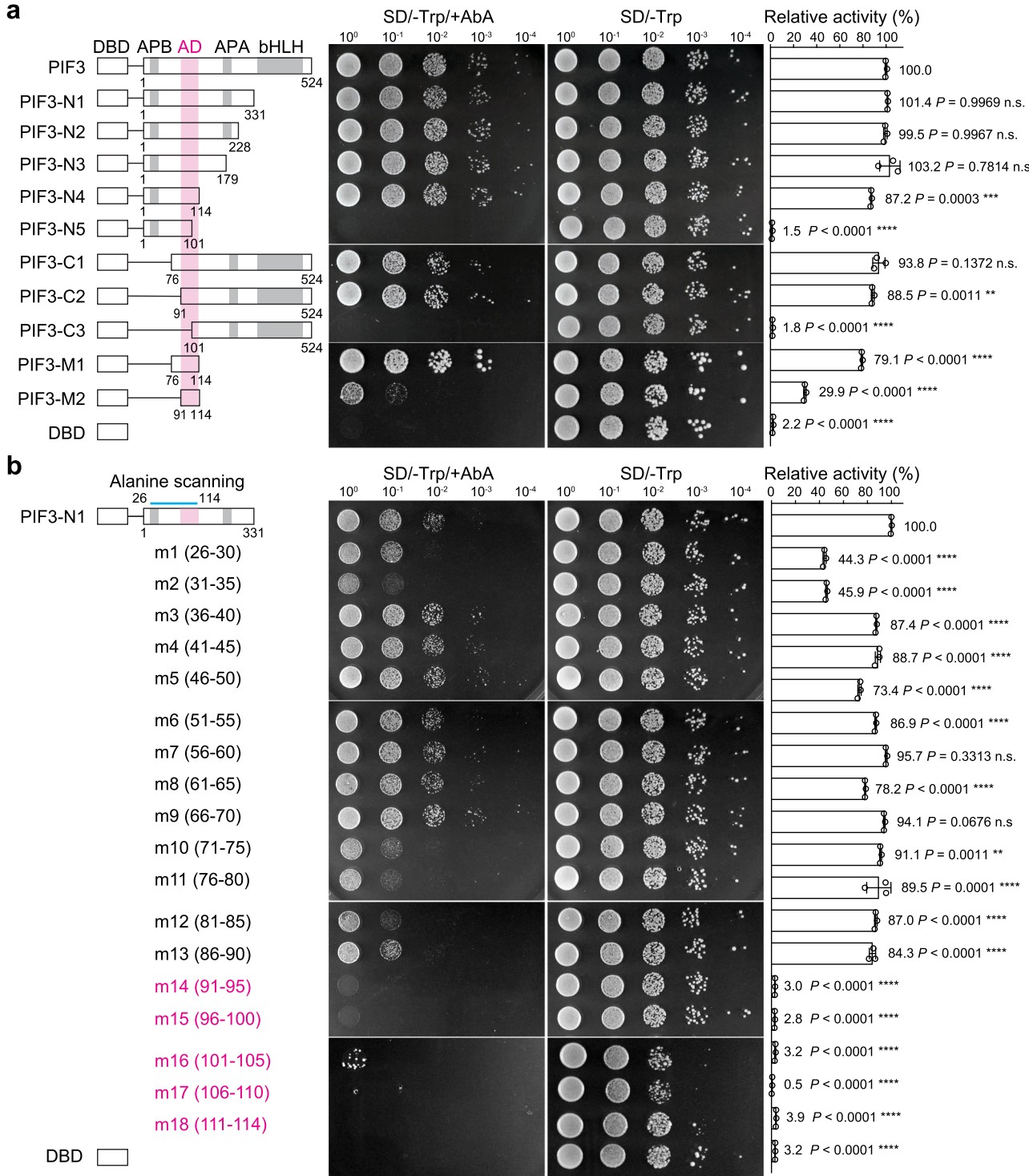

**Fig. 1 The aa$_{91-114}$ region confers PIF3's AD activity in yeast. a** Full-length and a series of truncation fragments of PIF3 were fused to the Gal4 DNA binding domain (DBD) as shown in the schematics, using the bait vector of the yeast two-hybrid system, and were evaluated for their transactivation activity in yeast. The magenta column highlights the aa$_{91-114}$ region. APB, active PHYB binding motif; APA, active PHYA binding motif. **b** Identification of the residues necessary for PIF3's AD activity via alanine scanning mutagenesis. A series of alanine-scanning mutants (m1–m18) were generated between amino acids 26 and 114 (indicated by the blue line) in the PIF3-N1 construct. The range of the alanine-substituted amino acids for each mutant is shown in parentheses. The mutants that lost their transactivation activity are shown in magenta. **a, b** The middle panel shows serial dilutions of the yeast strains containing the respective constructs grown on either SD/-Trp/+AbA or SD/-Trp (control) media. The right panel shows the relative transactivation activities quantified using the yeast liquid β-galactosidase assay. The transactivation activities were calculated relative to that of either PIF3 (**a**) or PIF3-N1 (**b**). DBD alone was used as a negative control. Error bars represent the s.d. of three biological replicates; the centers of the error bars represent the mean values. The numbers represent the mean value of the relative activity. The statistical significance of the changes between the sample and the control, PIF3 in **a** or PIF3-N1 in **b**, was analyzed by one-way ANOVA (Dunnett's test, *$P \leq 0.05$, **$P \leq 0.01$, ***$P \leq 0.001$, ****$P \leq 0.0001$, n.s. indicates no significant difference). The source data underlying the yeast liquid β-galactosidase assays in (**a**) and (**b**) are provided in the Source data file.

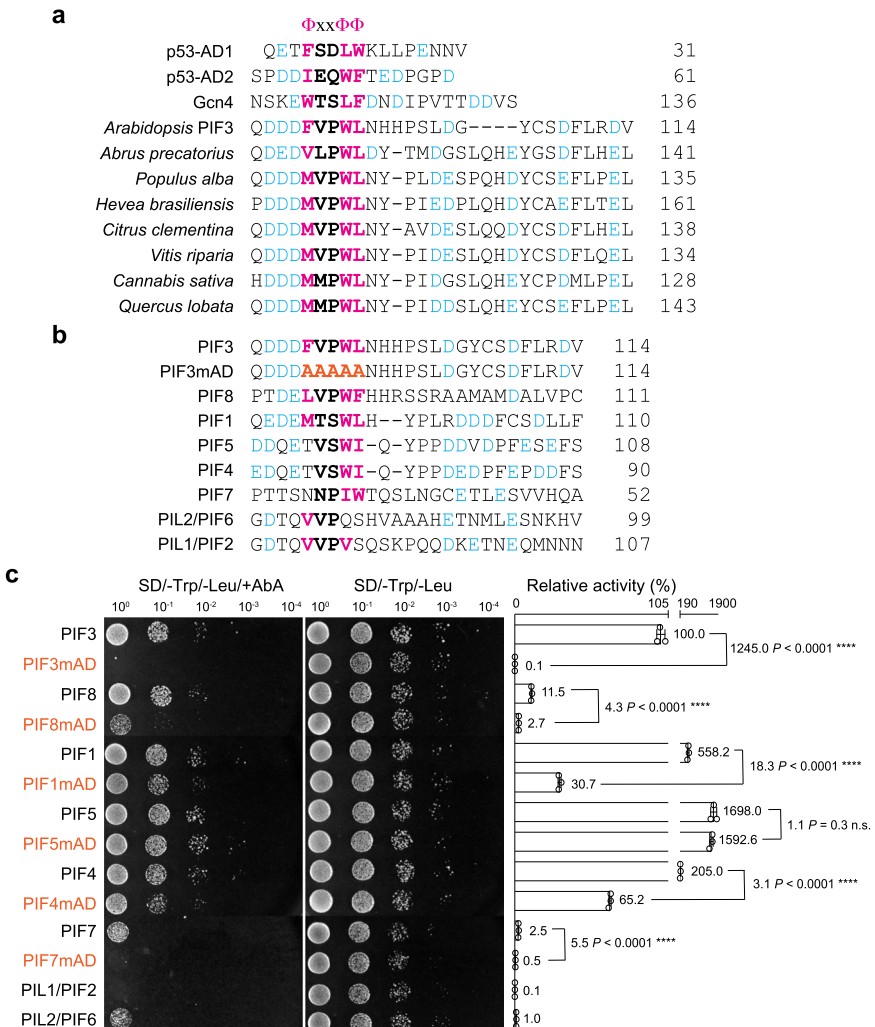

**Fig. 2 PIF3 possesses a p53-like AD. a** Amino acid sequence alignment of the ADs of *Homo sapiens* p53 (BAC16799.1), *Saccharomyces cerevisiae* Gcn4 (QHB08060.1), and select PIF3 orthologs from eudicots, including *Arabidopsis thaliana* (NP_001318964.1), *Abrus precatorius* (XP_027363600.1), *Populus alba* (XP_034896271.1), *Hevea brasiliensis* (XP_021639209.1), *Citrus clementina* (XP_006423962.1), *Vitis riparia* (XP_034707239.1), *Cannabis sativa* (XP_030504594.1), and *Quercus lobata* (XP_030957900.1). **b** Amino acid sequence alignment of the AD regions of PIF3, PIF3mAD, and the other PIF paralogs in *Arabidopsis*. The substituted alanines in PIF3mAD are labeled in orange. **a**, **b** The conserved activator ΦxxΦΦ motifs are highlighted in bold; the critical hydrophobic residues in the ΦxxΦΦ motif and the flanking acidic residues are labeled in magenta and blue, respectively. **c** Yeast transactivation assays showing the transactivation activities of the full-length PIF3 and PIF3 paralogs in *Arabidopsis* and the respective mAD mutants of PIF3, PIF8, PIF1, PIF5, PIF4, and PIF7. All mAD mutants contain five alanine substitutions in the predicted ΦxxΦΦ motif regions shown in (**b**). The left panel shows serial dilutions of the yeast strains containing the respective constructs grown on either SD/-Trp/-Leu/+AbA or SD/-Trp/-Leu (control) media. The right panel shows the relative transactivation activities quantified using the yeast liquid β-galactosidase assay. The transactivation activities were calculated relative to that of PIF3. Error bars represent the s.d. of three biological replicates; the centers of the error bars represent the mean values; the numbers represent the mean value of the relative activity. Fold changes in the activity of mAD mutants relative to their respective wild-type proteins are denoted. The statistical significance of the changes between the mAD mutants and their respective wild-type proteins was analyzed by two-tailed Student's *t*-test (****$P ≤ 0.0001$, n.s. indicates no significant difference). The source data underlying the yeast liquid β-galactosidase assays in (**c**) are provided in the Source data file.

clear transactivation activity (Fig. 2b, c). PIF1, PIF4, and PIF5 were the most potent activators, with significantly higher transactivation activity than PIF3, whereas PIF7 and PIF8 were less active than PIF3 (Fig. 2c). Conversely, PIL1 and PIL2, which show little similarity to the PIF3 AD in the corresponding region, had the least transactivation activity; PIL1 in particular showed no detectable activity (Fig. 2b, c). Interestingly, PIL1 has been suggested to heterodimerize with PIF4, PIF5, and PIF7 to negatively regulate their activities[55]. Therefore, the activity of the PIF family members might reflect their functional roles in PHY signaling.

To examine whether the ΦxxΦΦ motif contributes to the transactivation activities of PIF3 paralogs, we replaced the corresponding ΦxxΦΦ residues in PIF1, PIF4, PIF5, PIF7, and PIF8 with five alanines to create their respective mAD mutants. Intriguingly, only PIF7mAD lost its transactivation activity, suggesting the ΦxxΦΦ region in PIF7 is its sole AD (Fig. 2c). In contrast, mutating the ΦxxΦΦ motif did not abolish the activity of PIF1, PIF4, PIF5, or PIF8. PIF1mAD, PIF4mAD, and PIF8mAD remained active but had dramatically reduced activities compared with their respective wild-type proteins; the activity of PIF5 was barely affected by the mAD mutations (Fig. 2c). These results suggest that either the ΦxxΦΦ sequences in PIF1, PIF4, PIF5, and PIF8 are not essential for their AD activity or these PIFs contain one or more additional ADs. Therefore, neither the sequence nor the activity of the PIF3 AD is

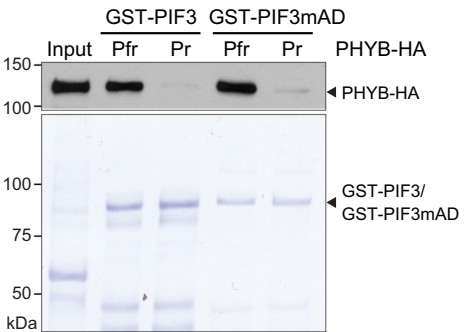
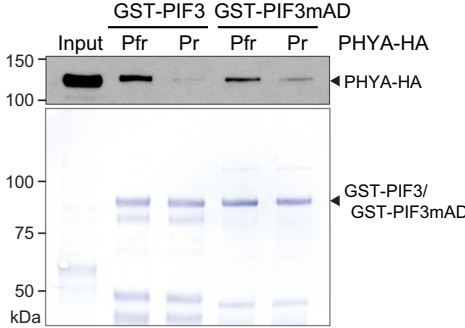

**Fig. 3 PIF3mAD retains light-dependent interactions with PHYA and PHYB.** GST pulldown results showing that PIF3mAD, like PIF3, interacts preferentially with active PHYA and PHYB. GST pulldown assays were performed using recombinant GST-PIF3, GST-PIF3mAD, or GST to pull down in vitro translated HA-tagged PHYA (PHYA-HA) or PHYB (PHYB-HA) conjugated with a phycocyanobilin chromophore. The pulldown assays were performed in either R or FR light to maintain PHYA-HA and PHYB-HA in the Pfr or Pr conformer, respectively. Input and bound PHYA-HA and PHYB-HA fractions were detected via immunoblots (upper panels) using anti-HA antibodies. Immobilized GST, GST-PIF3, and GST-PIF3mAD are shown in the Coomassie Brilliant Blue-stained SDS-PAGE gels (lower panels). The source data underlying the immunoblots and Coomassie Brilliant Blue-stained SDS-PAGE gels are provided in the Source data file.

perfectly conserved among PIF family members, suggesting a possible divergence in the mechanisms of transactivation among PIF members. These differences warrant further in-depth investigations on the AD functions of each individual PIF; we therefore focus the rest of the study on PIF3.

**PIF3mAD still binds PHYA and PHYB**. The AD of PIF3 lies between the APB and APA motifs (Fig. 1a)[24]. Next, we used in vitro GST pulldown assays to test whether the alanine mutations in PIF3mAD had an effect on PIF3 binding to PHYB and PHYA. Similar to GST-PIF3, GST-PIF3mAD could preferentially bind to the active Pfr forms of hemagglutinin (HA)-tagged PHYB and PHYA without a detectable change in affinity in the pull-down assays (Fig. 3). These results indicate that the FVPWL activator motif of PIF3 does not directly engage in the interactions with PHYA and PHYB.

**PIF3mAD is impaired in PIF3's functions in skotomorphogenesis**. To validate the PIF3 AD in vivo, we generated transgenic lines expressing HA- and YFP-tagged PIF3 (HA-YFP-PIF3) or PIF3mAD (HA-YFP-PIF3mAD) under the native PIF3 promoter as described previously[56]. The transactivation activity of PIF3 was best demonstrated during skotomorphogenesis. In dark-grown Arabidopsis seedlings, PIF3 binds to the enhancer regions of its target genes to activate their expression and consequently establish skotomorphogenesis, such as promoting hypocotyl elongation[28,57]. If the identified PIF3 AD is the sole functional AD in PIF3 in vivo, PIF3mAD should be defective in PIF3 functions in skotomorphogenesis. However, skotomorphogenesis is maintained redundantly by three additional PIF members, PIF1, PIF4, and PIF5, which hetero-dimerize or hetero-oligomerize with PIF3[21,28,30,58]. Therefore, the potential effects of PIF3mAD could be obscured in the presence of the other three PIFs. To circumvent this redundancy issue, we generated transgenic lines expressing HA-YFP-PIF3 or HA-YFP-PIF3mAD in a pif1pif3pif4pif5 quadruple mutant (pifq)[29]. For each construct, we picked two transgenic lines expressing similar levels of HA-YFP-PIF3 or HA-YFP-PIF3mAD. Despite being controlled by the native PIF3 promoter, the tagged PIF3 proteins in all four transgenic lines were expressed at higher levels than was endogenous PIF3 in the wild-type Col-0 and the pif145 triple mutant (Fig. 4a). We therefore assessed the activity of HA-YFP-PIF3mAD mainly via comparison with that of HA-YFP-PIF3. Dark-grown pifq seedlings display de-etiolated phenotypes with short hypocotyls compared with Col-0 (Fig. 4b, c)[29,30]. The two

transgenic lines expressing HA-YFP-PIF3 (PIF3/pifq 1-2 and 9-5) fully rescued the short-hypocotyl phenotype of pifq and had the same hypocotyl length as pif145, which expresses only PIF3 among the four PIFs (Fig. 4b, c). In contrast, the two lines expressing HA-YFP-PIF3mAD (PIF3mAD/pifq 2-1 and 4-5) only partially reverted the hypocotyl phenotype of pifq and remained significantly shorter than Col-0, pif145, and the two PIF3/pifq lines (Fig. 4b, c). HA-YFP-PIF3 and HA-YFP-PIF3mAD were expressed at similar levels in the transgenic lines, despite slightly less abundance of HA-YPF-PIF3mAD in the PIF3mAD/pifq lines (Fig. 4a). Both HA-YFP-PIF3 and HA-YFP-PIF3mAD localized to the nucleus (Fig. 4d). Therefore, the discrepancy in hypocotyl length between the PIF3/pifq and PIF3mAD/pifq lines was most likely due to the defect in the transactivation activity of HA-YFP-PIF3mAD. Indeed, the steady-state transcript levels of four PIF3 target genes[28], PIL1, ATHB-2, XTR7, and RD20, were significantly lower in the PIF3mAD/pifq lines than in the PIF3/pifq lines (Fig. 4e). Both the PIF3mAD/pifq and PIF3/pifq lines contained much higher levels of HA-YFP-tagged PIF3 compared with the endogenous PIF3 level in pif145 (Fig. 4a), however, the PIF3 target genes were mostly expressed at higher levels in the PIF3/pifq lines but at lower levels in the PIF3-mAD/pifq lines compared with pif145 (Fig. 4e). To exclude the less-likely possibility that the reduced expression of the PIF3 target genes in the PIF3mAD/pifq lines was due to a defect of PIF3mAD in DNA binding, we examined the binding of HA-YFP-PIF3 and HA-YFP-PIF3mAD to the G-box elements in the enhancer region of the marker gene PIL1 via chromatin immunoprecipitation (ChIP). The results of the ChIP experiments confirmed that the mutations in PIF3mAD did not affect its DNA binding activity (Fig. 4f). Together, these results support the conclusion that PIF3mAD is impaired in the transactivation activity of PIF3 and provide strong in vivo evidence validating the aa$_{91-114}$ region as PIF3's AD. The fact that the PIF3mAD/pifq lines could still partially reverse the hypocotyl phenotype of pifq (Fig. 4b, c) suggests that either the PIF3-mAD mutations do not completely abolish the AD activity in vivo, or alternatively, PIF3 can exert its functions through other associated transcription activators[56]. Supporting the latter possibility, genome-wide studies found that PIF-binding sites frequently coincide with binding sites of other families of transcription factors, some of which have been shown to directly interact with PIFs[59,60].

**PIF3 AD induces *ELIP2* activation by light**. PIF3 activates not only light-repressed genes but also, surprisingly, some light-inducible genes when dark-grown seedlings are exposed to light for the first time. For example, PIF3 participates in the light-

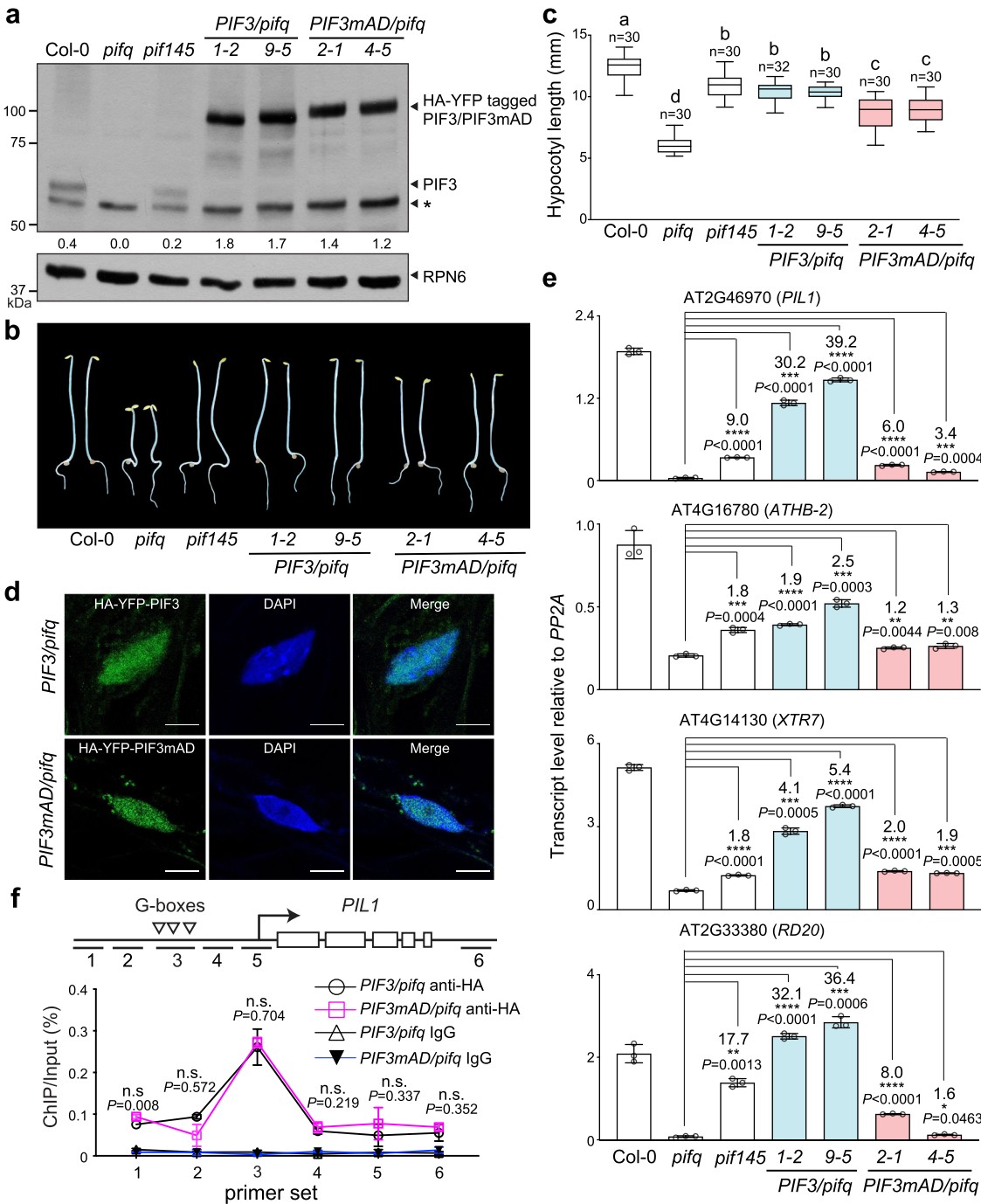

dependent induction of *ELIP2* (*EARLY LIGHT-INDUCED PROTEIN 2*), which encodes a special chlorophyll-binding protein that protects the plastids during initial exposure to light radiation[56,61]. However, it was unclear whether *ELIP2* induction relies on PIF3's intrinsic AD activity or the activity of other PIF3-associated transcription factors[56]. The PIF3mAD mutant therefore provides an opportunity to clarify the role of PIF3's intrinsic activity in *ELIP2* induction. To that end, first, we examined the involvement of the PIF3 AD in the early light-induced dynamics of PIF3 subnuclear distribution, phosphorylation, and abundance. During the dark-to-light transition, PHYA and PHYB recruit PIF3 to many small nuclear bodies and promote PIF3's phosphorylation and subsequent degradation[62,63]. These early light-signaling events depend on the interactions between PIF3 and PHYs[56,63]. Because PIF3mAD retains the binding activities to

active PHYA and PHYB (Fig. 3c), as expected, HA-YFP-PIF3mAD showed the same functionalities as HA-YFP-PIF3 in light-dependent nuclear body localization, phosphorylation, and degradation (Fig. 5a, b). However, in striking contrast, the *PIF3mAD/pifq* lines failed to activate *ELIP2* as effectively as did the *PIF3/pifq* lines (Fig. 5c). Within 1 h of R light exposure, the *PIF3/pifq* lines elevated the transcript level of *ELIP2* by more than 100 times, whereas the transcript level of *ELIP2* increased 4-to-5-fold less in the *PIF3mAD/pifq* lines (Fig. 5c). These results indicate that *ELIP2* activation is mainly induced by PIF3's intrinsic AD activity. Notably, *ELIP2* transcripts were also elevated significantly by light in *pifq*, albeit to a lesser extent (Fig. 5c), indicative of the involvement of other transcription factors in *ELIP2* induction besides the four PIFs. Therefore, PIF3 may promote the full induction of *ELIP2* through both its own AD as

**Fig. 4 PIF3mAD compromises PIF3's functions in skotomorphogenesis. a** Immunoblots showing the steady-state levels of endogenous PIF3, or HA-YFP-PIF3 and HA-YFP-PIF3mAD in dark-grown seedlings of Col-0, *pifq*, *pif145*, *PIF3/pifq* (*1-2* and *9-5*), and *PIF3mAD/pifq* (*2-1* and *4-5*) using anti-PIF3 antibodies. RPN6 was used as a loading control. The numbers beneath the PIF3 immunoblot indicate the relative levels of PIF3 or HA-YFP tagged PIF3 normalized to RPN6. The asterisk indicates non-specific bands. The immunoblot experiments were independently repeated at least three times, and the results of one representative experiment are shown. **b** Images of representative 4-d-old dark-grown seedlings of Col-0, *pifq*, and the transgenic *PIF3/pifq* and *PIF3mAD/pifq* lines. **c** Hypocotyl length measurements of the lines shown in (**b**). The boxes represent from the 25th to 75th percentile; the bars are equal to the median values; the whiskers extend to the maximum and minimum data points. Samples labeled with different letters exhibited statistically significant differences (ANOVA, Tukey's HSD, $P \leq 0.05$). **d** Confocal images showing the nuclear localization of HA-YFP-PIF3 and HA-YFP-PIF3mAD in hypocotyl epidermal cells of 4-d-old dark-grown *PIF3/pifq* (line *1-2*) and *PIF3mAD/pifq* (line *2-1*) seedlings. Nuclei were stained with DAPI. Scale bars represent 5 μm. **e** qRT-PCR results showing the steady-state transcript levels of select PIF3 target genes in seedlings described in (**b**). The transcript levels were calculated relative to those of *PP2A*. Error bars represent the s.d. of three biological replicates. The centers of the error bars represent the mean values. Numbers indicate fold changes relative to *pifq*; the statistical significance was analyzed using two-tailed Student's *t*-test (*$P \leq 0.05$, **$P \leq 0.01$, ***$P \leq 0.001$, ****$P \leq 0.0001$). **f** ChIP assays showing the binding of HA-YFP-PIF3 and HA-YFP-PIF3mAD to the G-box elements in the enhancer region of *PIL1*, as illustrated in the schematics. The ChIP experiments were performed using 4-d-old dark-grown *PIF3/pifq* (line *1-2*) and *PIF3mAD/pifq* (line *2-1*) seedlings with anti-HA antibodies. Immunoprecipitated DNA was quantified via real-time PCR using primer sets for locations 1 to 6 at the *PIL1* locus. ChIP with rabbit IgG was used as a negative control. Error bars represent the s.d. of three biological replicates. The centers of the error bars represent the mean values. The statistical significance was analyzed using two-tailed Student's *t*-test. n.s. indicates either the values between the *PIF3/pifq* and *PIF3mAD/pifq* samples are not significantly different or the difference is <1.5-fold. The source data underlying the immunoblots in (**a**), hypocotyl measurements in (**b**), qRT-PCR data in (**e**), and ChIP data in (**f**) are provided in the Source data file.

well as the activity of its associated transcription factors[56,60], which also provides an explanation for the residual light response of *ELIP2* in the *PIF3mAD/pifq* lines (Fig. 5c).

**PIF3 AD participates in gene activation by shade.** PIF3 plays an important role in the rapid gene activation when PHYB is inactivated by shade or FR light[64]. To assess the function of the PIF3 AD in the activation of shade-inducible genes during the light-to-shade transition, first, we characterized *PIF3/pifq* and *PIF3mAD/pifq* lines grown in continuous R light. Interestingly, while *pifq* shows a short-hypocotyl phenotype in the light, the slow growth of *pifq* was fully rescued in the *PIF3/pifq* lines (Fig. 6a, b). Under R light conditions, endogenous PIF3 does not accumulate to a detectable level. However, like the previously reported *PIF3* transgenic lines[33], the two *PIF3/pifq* lines accumulated a significant amount of HA-YFP-PIF3 in the light (Fig. 6c), providing an explanation for their enhanced hypocotyl elongation in the light. On the other hand, the *PIF3mAD/pifq* lines only partially reversed the dwarf phenotype of *pifq*, and they were significantly shorter than the *PIF3/pifq* lines (Fig. 6a, b). HA-YFP-PIF3mAD accumulated to similar levels as HA-YFP-PIF3 (Fig. 6c), but the transcript levels of three PIF3-target genes, *PIL1*, *ATHB-2*, and *IAA29*, in *PIF3mAD/pifq* were similar to those in *pifq* and failed to increase to the levels in Col-0 and *PIF3/pifq* (Fig. 6d and Supplementary Fig. 3). Next, we treated the R-light-grown seedlings with 1 h of FR light (simulated shade) and then examined the expression of the same three PIF target genes before and after the FR treatment. The *PIF3/pifq* lines were able to dramatically enhance the expression of PIF3 target genes as Col-0 (Fig. 6d). However, like *pifq*, the *PIF3mAD/pifq* lines failed to activate PIF3 target genes to the extend as those in Col-0 (Fig. 6d), indicating that the PIF3 AD is required for the full activation of shade-inducible genes. It is important to note that our experimental design was not aimed at assessing the contribution of PIF3 in shade responses; because PIF3 accumulates to higher levels in our transgenic lines, the contribution of PIF3 to shade responses in our experimental settings was likely overestimated. However, our results clearly show that PIF3mAD is impaired in the transactivation activity under shade treatment.

**Binding of PHYB and PHYA inhibits the transactivation activity of PIF3 and PIF1.** In the shade experiments, we were intrigued by the fact that although HA-YFP-PIF3 accumulated to

high levels in the *PIF3/pifq* lines (Fig. 6c), the transcript levels of the PIF3 target genes remained low in continuous R light (Fig. 6d). These observations suggest that the transactivation activity of HA-YFP-PIF3 must be repressed in the light by PHYB, and likely, this transcriptional repression is released by FR light upon PHYB inactivation. It has been postulated that PHYB inhibits PIF3 binding to the promoters of target genes[41,43]. However, it remains ambiguous whether PHYB influences the DNA-binding activity of PIF3 directly or indirectly through the regulation of PIF3's AD activity. Because PIF3 AD resides in close proximity to the APB motif, we hypothesized that binding of PHYB to the APB could directly inhibit the AD activity of PIF3 independently of its DNA-binding activity. To discern the AD activity of PIF3 from its DNA binding, we went back to the yeast transactivation assay and examined the activity of the PIF3-N4 fragment, which includes only the PIF3 AD and the APB motif without the bHLH DNA-binding domain (Fig. 7a). We coexpressed PIF3-N4 with PHYB conjugated with a phycocyanobilin (PCB) chromophore and performed yeast liquid β-galactosidase assays in either R or FR light to test whether the activity of PIF3-N4 could be influenced by the Pfr or the Pr form of PHYB, respectively (Fig. 7a). We found that the Pfr form of PHYB reduced the activity of PIF3-N4 by more than 70%, whereas the Pr form imposed little effect on PIF3-N4's activity (Fig. 7b). Because PIF3 interacts with both PHYB's N- and C-terminal modules[65], we then tested which interaction is responsible for the transcriptional inhibition. For the N-terminal module of PHYB, we adopted a previously demonstrated biologically active form called NGB, which combines the N-terminal module of PHYB with GFP and GUS as a dimerization domain[17]. NGB was also able to inhibit the transactivation activity of PIF3-N4 in a light-dependent manner; in contrast, the C-terminal module of PHYB had little effect (Fig. 7b). These results demonstrate that the N-terminal photosensory module of PHYB can inhibit the transactivation activity of PIF3's AD, and this action of PHYB is independent of PIF3's DNA-binding activity. Interestingly, PHYB and NGB failed to repress the activity of PIF3-M1, which contains the PIF3 AD but not the APB motif (Fig. 7c), supporting the notion that the photoinhibition of the PIF3 AD is mediated by a direct interaction between PHYB and the APB motif. Both PHYB and NGB could also repress the activity of PIF1 in a light-dependent manner (Fig. 7d), suggesting that PHYB can inhibit the activity of other PIFs via the same mechanism. Besides the APB motif, PIF3 and PIF1 contain the APA motif near the AD

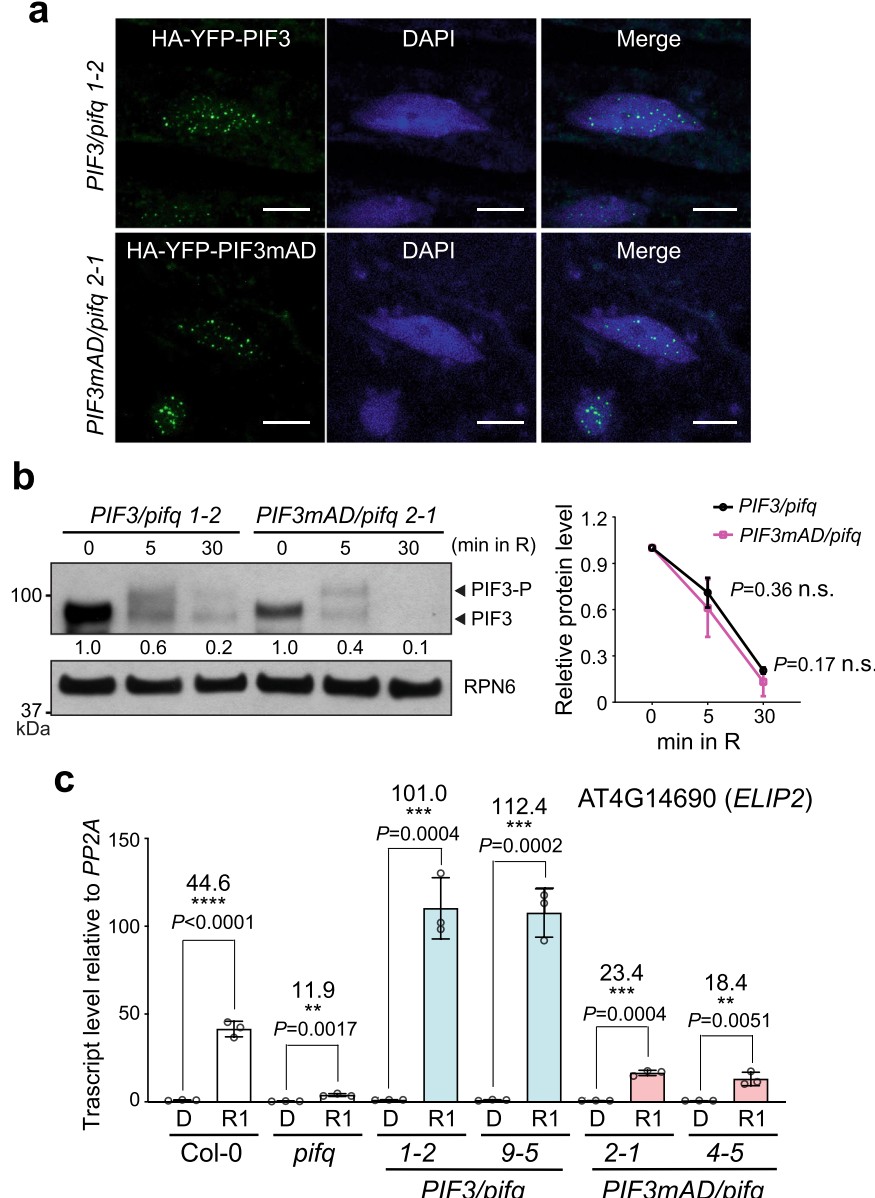

**Fig. 5 Characterization of PIF3mAD in early light signaling. a** Confocal images showing the nuclear body localization of HA-YFP-PIF3 and HA-YFP-PIF3mAD in the hypocotyl epidermal cells of dark-grown *PIF3/pifq* (line *1-2*) and *PIF3mAD/pifq* (line *2-1*) seedlings after 1 min of exposure to 10 μmol m$^{-2}$ s$^{-1}$ R light. Nuclei were stained with DAPI. The scale bars represent 5 μm. The localization patterns of HA-YFP-PIF3 and HA-YFP-PIF3mAD were observed in at least five individual seedlings of each line. **b** Immunoblots showing the phosphorylation and degradation of HA-YFP-PIF3 and HA-YFP-PIF3mAD in 4-d-old *PIF3/pifq* and *PIF3mAD/pifq* seedlings during the dark-to-light transition at the indicated time points in 10 μmol m$^{-2}$ s$^{-1}$ R light. HA-YFP-PIF3 and HA-YFP-PIF3mAD were detected using anti-HA antibodies. RPN6 was used as a loading control. The immunoblot experiments were independently repeated four times, and the immunoblots of one representative experiment are shown. The numbers beneath the PIF3 immunoblot indicate the relative levels of HA-YFP-tagged PIF3 or PIF3mAD relative to the abundance of each line in the darkness. The right panel shows the kinetic changes of HA-YFP-PIF3 and HA-YFP-PIF3mAD during the dark-to-light transition using data from all four experiments. The protein levels of PIF3 and PIF3mAD were quantified relative to those at time 0. Although *PIF3mAD/pifq* accumulated slightly less HA-YFP-PIF3mAD than HA-YFP-PIF3 in *PIF3/pifq* in darkness, the degradation kinetics of HA-YFP-PIF3mAD was the same as that of HA-YFP-PIF3. Error bars represent the s.d. of four biological replicates. The centers of the error bars represent the mean values. The statistical significance was analyzed using two-tailed Student's *t*-test. n.s. indicates the values between the *PIF3/pifq* and *PIF3mAD/pifq* samples are not significantly different. **c** qRT-PCR results showing the steady-state transcript levels of *ELIP2* in seedlings of 4-d-old dark-grown Col-0, *pifq*, and the transgenic *PIF3/pifq* and *PIF3mAD/pifq* lines before (D) and 1 h after (R1) exposure to 10 μmol m$^{-2}$ s$^{-1}$ R light. The transcript levels were calculated relative to those of *PP2A*. The numbers represent fold changes of the transcript level of *ELIP2* in R1h compared to that in D. Error bars represent the s.d. of three biological replicates. The centers of the error bars represent the mean values. The statistical significance was analyzed using two-tailed Student's *t*-test (**$P \leq 0.01$, ***$P \leq 0.001$, ****$P \leq 0.0001$). The source data underlying the immunoblots in (**b**) and qRT-PCR data in (**c**) are provided in the Source data file.

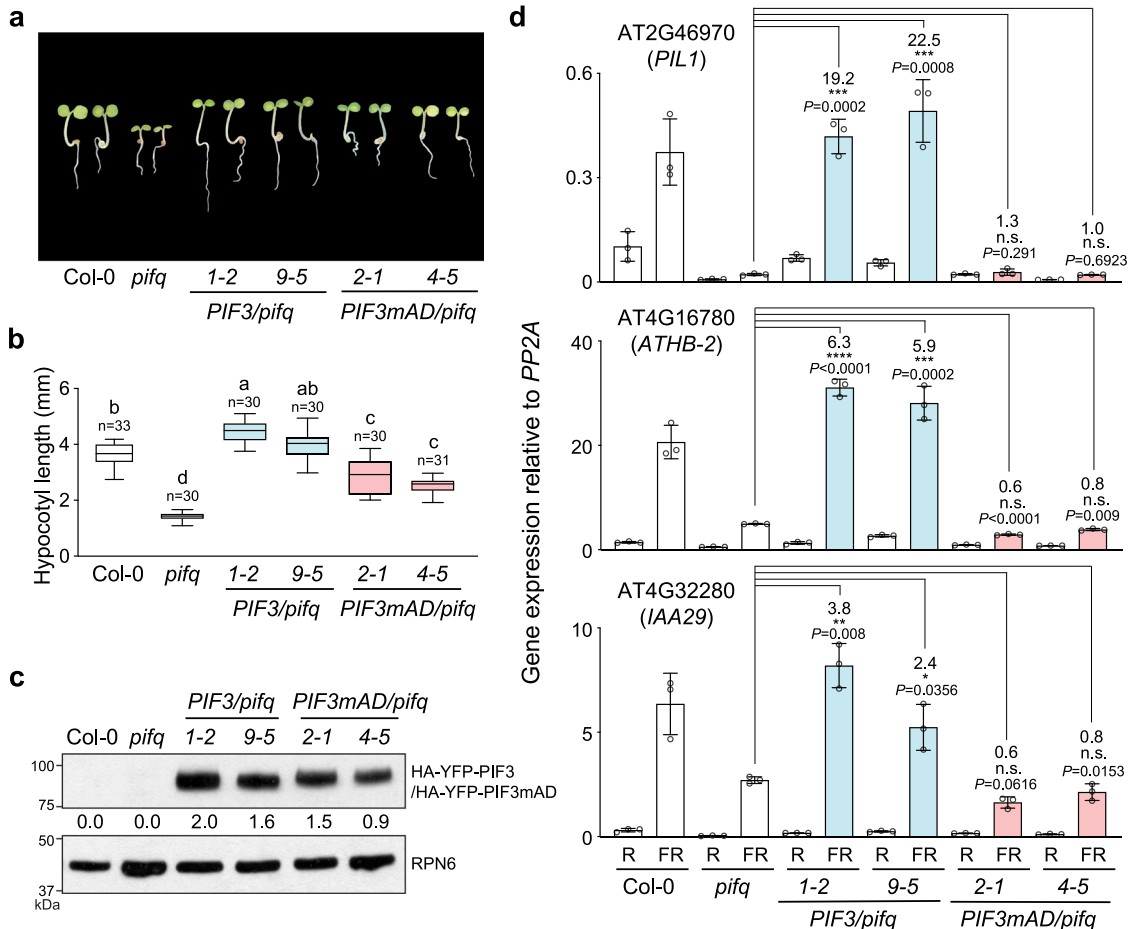

**Fig. 6 PIF3's AD participates in gene activation by shade. a** Images of representative 4-d-old seedlings of Col-0, *pifq*, two *PIF3/pifq* lines, and two *PIF3mAD/pifq* lines grown in 10 μmol m$^{-2}$ s$^{-1}$ R light. **b** Hypocotyl length measurements of the lines shown in (**a**). The boxes represent the 25th to 75th percentiles; the bars are equal to the median values; the whiskers extend to the maximum and minimum data points. Samples labeled with different letters exhibited statistically significant differences in hypocotyl length (ANOVA, Tukey's HSD, $P \leq 0.05$). **c** Immunoblots showing the steady-state levels of HA-YFP-PIF3 or HA-YFP-PIF3mAD in the *PIF3/pifq* and *PIF3mAD/pifq* seedlings shown in (**a**) using anti-HA antibodies. RPN6 was used as a loading control. The immunoblot experiments were independently repeated at least three times, and the results of one representative experiment are shown. The numbers beneath the immunoblot indicate the relative levels of HA-YFP-PIF3 or HA-YFP-PIF3mAD normalized to RPN6. **d** qRT-PCR results showing the steady-state transcript levels of select PIF3 target genes in 4-d-old R-light-grown seedlings of Col-0, *pifq*, and the *PIF3/pifq* and *PIF3mAD/pifq* lines before (R) and after a 1-h FR treatment (FR). The transcript levels were calculated relative to those of *PP2A*. Error bars represent the s.d. of three biological replicates. The centers of the error bars represent the mean values. Numbers indicate fold changes relative to *pifq*; the statistical significance was analyzed using two-tailed Student's *t*-test (*$P \leq 0.05$, **$P \leq 0.01$, ***$P \leq 0.001$, ****$P \leq 0.0001$); n.s. indicates the difference was either <2-fold or not statistically significant. The source data underlying the immunoblots in (**c**) and qRT-PCR data in (**d**) are provided in the Source data file.

for PHYA binding. Interestingly, coexpressing PHYA in yeast also inhibited the transactivation activity of PIF3 and PIF1 in a light-dependent manner (Fig. 7e), suggesting binding of PHYA to APA can also inhibit the AD activity.

**NGB represses PIF3 activity via the light-sensing knot.** Previous studies have identified three mutations in PHYB's light-sensing knot, R110Q, G111D, and R352K, which disrupt the PHYB-APB interaction[66,67]. We examined whether these mutations could attenuate the effect of NGB on PIF3's activity. Indeed, all three mutations in the light-sensing knot significantly reduced the inhibitory function of NGB on PIF3's AD activity (Fig. 8a). NGB can rescue the long hypocotyl phenotype of the null *phyB-5* mutant[17]. The in vivo activity of NGB relies on its interaction with the APB motif in PIFs, as NGB-R110Q, NGB-G111D, and NGB-R352K failed to rescue *phyB-5*[66,67]. Because NGB has little activity in promoting PIF3 degradation, it was proposed that NGB regulates hypocotyl growth by inhibiting the function of

PIF3[16,41–43]. Our new findings raised the possibility that NGB controls hypocotyl growth by directly repressing the AD activity of PIF3. To test this hypothesis, we grew *NGB, NGB-R110Q, NGB-G111D,* and *NGB-R352K* seedlings in continuous R light and compared the protein levels of PIF3 as well as the transcript levels of select PIF3 target genes. PIF3 accumulated to similar levels in *NGB, NGB-R110Q, NGB-G111D, NGB-R352K,* and *phyB-5*, confirming that NGB, as well as the NGB mutants, have little activity in promoting PIF3 degradation in the light (Fig. 8b)[16,41–43]. However, despite the similar levels of PIF3, the transcript levels of PIF3 target genes in *NGB* remained low and were similar to those in *PBG*, in which PIF3 was undetectable (Fig. 8b, c), suggesting that NGB can almost fully block the activity of PIF3 in vivo. The transcriptional repression activity of NGB was greatly compromised in *NGB-R110Q, NGB-G111D,* and *NGB-R352K*, as all three mutants showed elevated transcript levels of PIF3 target genes similar to those in *phyB-5* (Fig. 8c), indicating that disrupting the PHYB-APB interaction almost completely abolished the transcriptional repression activity of

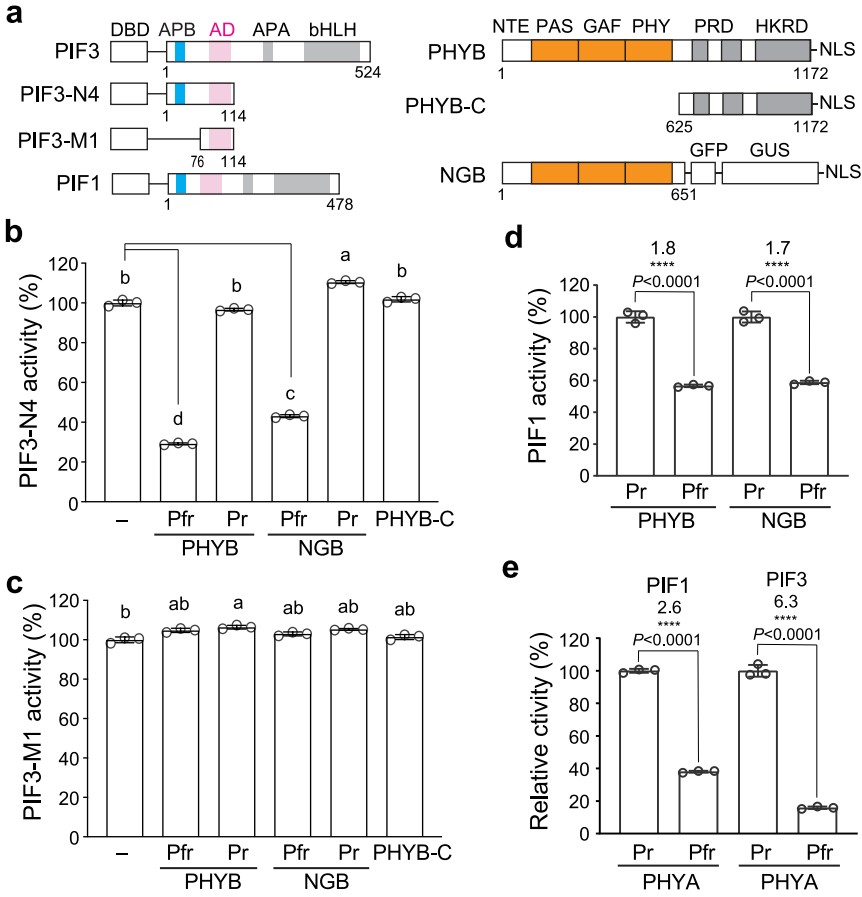

**Fig. 7 Binding of either PHYB or PHYA inhibits the AD activity of PIF3 and PIF1. a** Schematics of the PIF3, PIF1, and PHYB constructs used for determining the effect of PHYB binding to the transactivation activity of PIF3 and PIF1. NLS, nuclear localization signal. **b** Results of yeast liquid β-galactosidase assays showing the activity of PIF3-N4 expressed alone (-) or coexpressed with PHYB, NGB, or PHYB-C. **c** Results of yeast liquid β-galactosidase assays showing the activity of PIF3-M1 expressed alone (-) or coexpressed with PHYB, NGB, or PHYB-C. For **b** and **c**, The assays for PHYB and NGB were performed in either R or FR light to maintain PHYB and NGB in the Pfr or the Pr conformer, respectively. The transactivation activities were calculated relative to that of the yeast strains expressing the PIF3 fragments alone. Error bars represent the s.d. of three biological replicates. Samples labeled with different letters exhibited statistically significant differences (ANOVA, Tukey's HSD, $P \leq 0.05$, $n = 3$). **d** Results of yeast liquid β-galactosidase assays showing the activity of PIF1 coexpressed with PHYB or NGB. **e** Results of yeast liquid β-galactosidase assays showing the activity of PIF1 or PIF3 coexpressed with PHYA. For **d** and **e**, the assays for PHYB, NGB, and PHYA were performed in either R or FR light to maintain them in the Pfr or the Pr conformer, respectively. The transactivation activities were calculated relative to the assays when PHYs were in the Pr. The statistical significance was analyzed using two-tailed Student's t-test (****$P \leq 0.001$); The numbers indicate fold changes in activity. Error bars represent the s.d. of three biological replicates. The centers of the error bars represent the mean values. The source data underlying the yeast liquid β-galactosidase assays in (**b**), (**c**), (**d**), and (**e**) are provided in the Source data file.

NGB. These results thus provide genetic evidence supporting the model that binding of PHYB's light-sensing knot to the APB motif of PIF3 directly represses the transactivation activity of the PIF3 AD in vivo.

## Discussion

The PHYB-PIF signaling module is at the center of plant-environment interactions that controls the expression of hundreds of light-responsive genes, thereby profoundly influencing almost all aspects of plant development, growth, metabolism, and immunity[22,26]. However, the AD sequences of PIFs and the regulatory mechanism of PIFs' intrinsic AD activity had not been precisely determined. This study defines a 24-amino-acid AD in PIF3 and identifies the functional attributes of the PIF3 AD including an essential ΦxxΦΦ activator motif and flanking acidic residues, which strongly resemble the ADs of the mammalian tumor suppressor p53 and the yeast activator Gcn4[47,48,50,51]. These findings reveal the unexpected conservation of sequence-specific ADs across the animal, fungal, and plant kingdoms.

Moreover, we uncovered a novel PHYB signaling mechanism via direct photoinhibition of the AD activity of PIF3 (Fig. 9). These results, combined with those of previously published studies on the structure–function relationship of PHYB[16,41–43], indicate that the transactivation activity and stability of PIF3 are controlled by structurally separable mechanisms via PHYB's N-terminal photosensory and C-terminal output modules, respectively (Fig. 9). While the light-independent interaction between the C-terminal module of PHYB and PIF3 promotes PIF3 degradation, the light-reversible interaction between the N-terminal photosensory module of PHYB and the APB of PIF3 directly modulates PIF3's transactivation activity (Fig. 9). We propose that PHYB, and likely PHYA, regulates the stability and activity of other PIFs also via the dual mechanisms.

We demonstrated that PIF3 possesses a single AD in the $aa_{91-114}$ region between the APB and APA motifs. The PIF3 AD defined by this study is consistent with the previously identified activating region in yeast between amino acids 90 and 120[49]. We further narrowed down the AD region and, more importantly,

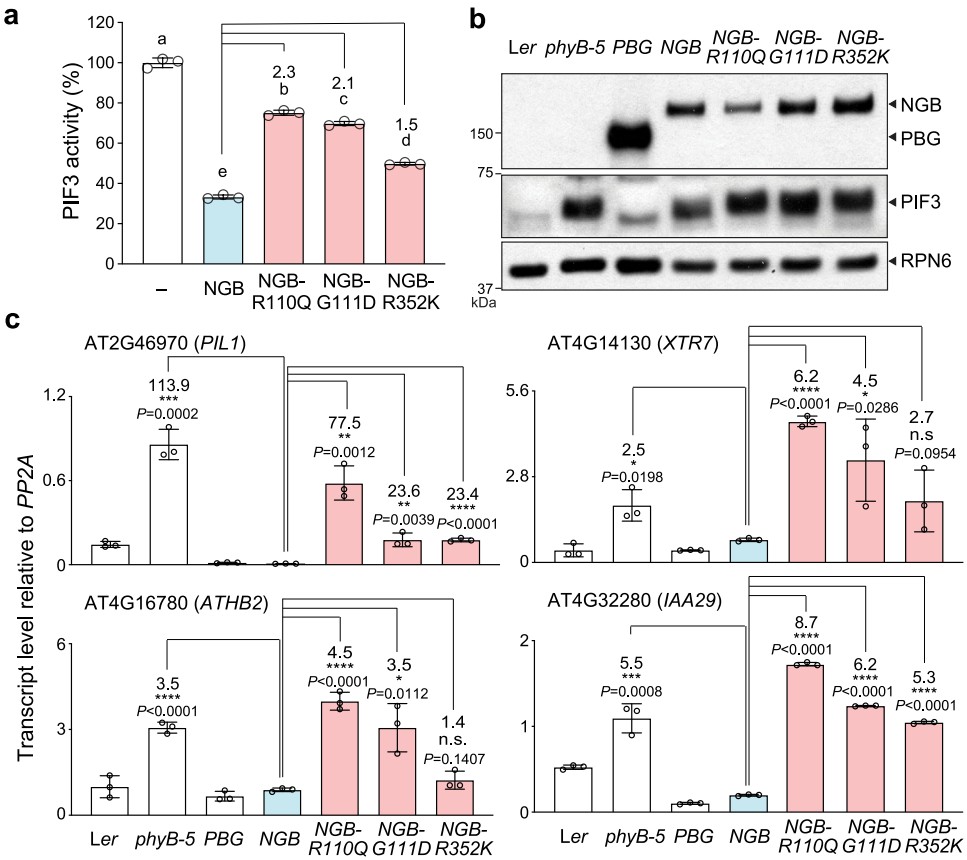

**Fig. 8 NGB inhibits PIF3 activity via light-sensing knot. a** Results of yeast liquid β-galactosidase assays showing the activity of PIF3 expressed alone (-) or coexpressed with NGB or NGB mutants. The transactivation activities were calculated relative to that of the yeast strains expressing PIF3 alone. Samples labeled with different letters exhibited statistically significant differences in transactivation activity (ANOVA, Tukey's HSD, $P < 0.05$, $n = 3$). The numbers represent fold-changes between the NGB mutants and NGB. Error bars represent the s.d. of three biological replicates. The centers of the error bars represent the mean values. **b** Immunoblots showing the steady-state levels of PBG, NGB, and PIF3 in 4-d-old L*er*, *phyB-5*, *PBG*, *NGB*, *NGB-R110Q*, *NGB-G111D*, and *NGB-R352K* seedlings grown in 10 μmol m$^{-2}$ s$^{-1}$ R light. PBG and NGB were detected using anti-GFP antibodies, and PIF3 was detected using anti-PIF3 antibodies. RPN6 was used as a loading control. The immunoblot experiments were independently repeated at least three times, and the results of one representative experiment are shown. **c** qRT-PCR results showing the steady-state transcript levels of PIF3 target genes, *PIL1*, *XTR7*, *ATHB2*, and *IAA29*, in the lines described in (**b**). The transcript levels were calculated relative to those of *PP2A*. The numbers indicate fold changes relative to *NGB*; the statistical significance was analyzed using two-tailed Student's *t*-test (*$P \leq 0.05$, **$P \leq 0.01$, ***$P \leq 0.001$, ****$P \leq 0.0001$). Error bars represent the s.d. of three biological replicates. The centers of the error bars represent the mean values. The source data underlying the yeast liquid β-galactosidase assays in (**a**), the immunoblots in (**b**), and the qRT-PCR data in (**c**) are provided in the Source data file.

showed that this is the only AD in PIF3, revealed its similarities to the ADs of p53 and Gcn4, validated the function of the PIF3 AD in vivo, and uncovered the direct regulation of the PIF3 AD by PHYB. The conclusion that PIF3 possesses a single p53-like AD is supported by both transactivation assays in yeast and the characterization of the PIF3mAD mutant in planta. First, truncation analysis and alanine-scanning mutagenesis of PIF3 demonstrated that aa$_{91-114}$ is the only region necessary and sufficient for PIF3's AD activity in yeast (Fig. 1 and Supplementary Fig. 1). Different from the prior work, which suggested a second AD between amino acids 27 and 43, aa$_{27-43}$ did not show any AD activity in our experiments (Fig. 1 and Supplementary Fig. 1). Second, the PIF3 AD shares similar attributes as the ADs of p53 and Gcn4, including an ΦxxΦΦ activator motif and flanking acidic residues (Fig. 2a). Third, the PIF3mAD mutant in which the ΦxxΦΦ activator motif is replaced with five alanines compromises the transactivation functions of PIF3 in both yeast and *Arabidopsis* (Figs. 2, 4, 5, and 6), providing strong evidence validating the function of the PIF3 AD. The PIF3 AD is highly conserved in PIF3 orthologs in eudicots, suggesting evolutionarily-conserved roles in PIF3's activator function (Fig. 2a).

Our results unveil a novel mechanism of PHYB signaling via direct photoinhibition of the transactivation activity of PIF3. The PIF3 AD resides between the APB and APA motifs (Fig. 9)[24,66,67]. We showed that the light-induced interaction between the N-terminal photosensory module of PHYB and the APB repressed the transactivation activity of PIF3 in yeast, whereas the C-terminal output module of PHYB had no effect on PIF3's activity (Fig. 7b, c). Consistent with the conclusion, the inhibitory effect of PHYB on PIF3's activity was significantly attenuated when the PHYB-APB interaction was disrupted by the R110Q, G111D, and R352K mutations in the light-sensing knot of PHYB (Fig. 8a)[66,67]. Furthermore, Using NGB, which contains only the N-terminal module of PHYB, as a model, we demonstrated that the interaction between PHYB's photosensory module and the APB motif of PIF3 is required for repressing the transactivation activity of PIF3 in vivo (Fig. 8b, c)[17,41–43]. These results thus uncover a new signaling mechanism by the photosensory module of PHYB via direct inhibition of PIF3's AD activity, possibly by blocking the AD from binding to transcriptional coactivators. Our results confirmed that NGB or the N-terminal module of PHYB is not responsible for promoting PIF3 degradation

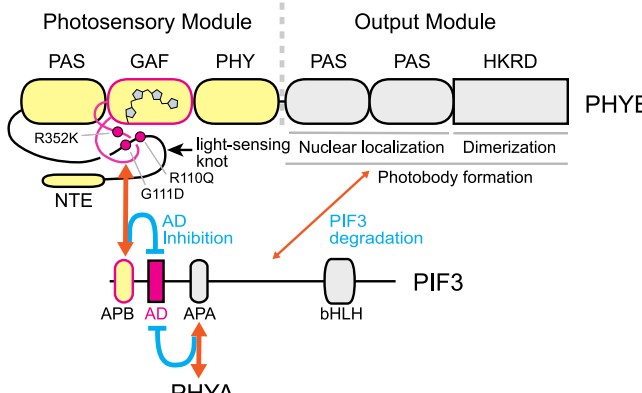

**Fig. 9 Model of two structurally separable PHYB actions for inhibiting PIF3 stability and transactivation activity.** Photoactivation of the N-terminal photosensory module induces the binding of PHYB's light-sensing knot to PIF3's APB, thereby masking the transactivation activity of the adjacent PIF3 AD. In parallel, the C-terminal output module, which mediates photobody formation, interacts weakly with PIF3 in a light-independent manner to promote PIF3 degradation. The latter action requires the dimerization of the output module via HKRD. The orange lines represent the interactions between PHYB and PIF3. Binding of PHYA to APA also inhibits the activity of the PIF3 AD. The schematic of PHYB domain structure is modified from a model by Burgie and Viestra[12].

(Fig. 8b)[41,42]. Consistent with this conclusion, it was shown previously that a constitutively active version of the photosensory module of PHYB, which harbors the Y276H mutation, also failed to trigger PIF3 degradation in the light[68]. Together, these results, combined with previously published data on the function of PHYB's C-terminal output module in PIF3 degradation[16], indicate that PHYB regulates the activity and stability of PIF3 through structurally separable actions of the N-terminal photosensory module and C-terminal output module, respectively (Fig. 9). Because the PIF3 AD is conserved in other PIF members including PIF1, PIF4, PIF5, PIF7, and PIF8 (Fig. 2b, c), it is likely that PHYB also controls these PIFs via the dual mechanisms. Supporting this notion, the photosensory module of PHYB can inhibit the activity of PIF1 in a light-dependent manner in yeast (Fig. 7d). Moreover, our results showed that photoactivated PHYA could inhibit the activity PIF3 and PIF1 in yeast (Fig. 7e), suggesting that PHYA may also transduce light signals through direct inhibition of the AD activity of PIF3 and PIF1. Compared with the mechanism of PIF degradation, direct inhibition of PIFs' transactivation activity provides a rapid mode of photoresponsive gene regulation, and also, it is likely a critical mechanism to regulate the PIFs that can accumulate in the light, such as PIF4, PIF5, and PIF7[36–39].

PHYs trigger rapid phosphorylation of PIF3 by photoregulatory protein kinases (PPKs) in the light[34,63]. The PIF3 AD contains two light-dependent phosphorylation sites, S102 and S108[69]. Phosphorylation at these sites is expected to increase negative charges around the activator motif, which could potentially alter the accessibility of the activator motif and/or the affinity of the AD to transcriptional coactivators. Therefore, PHYB and PHYA may also modulate the activity of the PIF3 AD through phosphorylation. This hypothesis needs to be further examined in future studies.

The interaction between PHYB and PIF3 was shown to reduce PIF3's DNA-binding activity or enhance PIF3 sequestration away from target gene promoters independently of PHYB-mediated PIF3 degradation[41,43]. The sequestration activity of PHYB relies only on PHYB's N-terminal module and is abolished by the

G111D mutation in the light-sensing knot[41,43]. Our results here thus raised the question of the relationship between PIF3's transactivation activity and DNA binding. To discern PHYB's function in inhibiting the PIF3 AD from its role in regulating PIF3 DNA-binding activity, we chose to use the PIF3-N4 fragment to assess the photoinhibition of PIF3's AD by PHYB, because PIF3-N4 does not contain the bHLH domain, and therefore, its DNA binding in yeast relies on the Gal4-DBD instead (Fig. 7a, b). These experiments demonstrate that the N-terminal module of PHYB can directly inhibit the transactivation activity PIF3, independently of PIF3's bHLH activity—i.e., the photoinhibition of the activity of the PIF3 AD by PHYB is not a consequence of a reduction in PIF3's DNA-binding activity. It is conceivable that blocking the activity of PIF3's AD by PHYB may disrupt the interactions of PIF3 to transcriptional coactivators, thereby reducing PIF3's association to target gene promoters. Alternatively, it is equally possible that, in addition to repressing the transactivation activity, binding of PHYB to the APB motif of PIF3 also triggers conformational changes that allosterically reduce the DNA-binding activity of the bHLH domain. The current data cannot distinguish between these two models.

It was demonstrated more than thirty years ago that a transcriptional activator from one species, such as the yeast activator Gal4 and the herpes simplex virus protein VP16, can act as a potent activator in animal, yeast, and plant cells, leading to the notion that the basic mechanism of transcription activation is likely conserved in eukaryotes[44,70,71]. However, it has been puzzling why transcription can be activated by diverse AD sequences across eukaryotic kingdoms. The identification of the sequence-specific ΦxxΦΦ motif in animal and yeast activators suggests that the mechanism of transcriptional activation could be conserved at the primary sequence level and the intrinsic disordered property of ADs[47,48,50,51]. To our knowledge, the PIF3 AD is the first sequence-specific AD identified in plants. Our results therefore provide initial evidence supporting the conservation of sequence-specific ADs across the animal/fungal and plant kingdoms.

In conclusion, our results identify a p53-like AD in the PIF family of transcriptional activators and unveil a novel light signaling mechanism via direct photoinhibition of PIFs' transactivation activity by PHYB and likely PHYA. It will be of interest in future work to investigate the transactivation mechanism by PIFs' ADs and how binding of PHYB inhibits such mechanism.

## Methods

**Plant materials, growth conditions, and hypocotyl measurement**. *Arabidopsis* wild-type Columbia (Col-0) and Landsberg *erecta* (L*er*), as well as the *pifq*[29] (Col-0), *pif145*[29], and *phyB-5*[72](L*er*) mutants, were used as controls to characterize hypocotyl growth and gene expression under various light conditions. The *PBG*, *NGB*, *NGB-R110Q*, *NGB-G111D*, and *NGB-R352K* lines have been previously reported[17,66]. The *PIF3/pifq* (*1-2* and *9-5*) and *PIF3mAD/pifq* (*2-1* and *4-5*) transgenic lines were generated in this study.

*Arabidopsis* seeds were surface-sterilized and plated on half-strength Murashige and Skoog medium containing Gamborg's vitamins (Caisson Laboratories), 0.5 mM MES pH 5.7, and 0.8% agar (w/v). Seeds were stratified in the dark at 4 °C for 5 days and grown at 21 °C in an LED chamber (Percival Scientific, Perry, IA) in the indicated light conditions. Fluence rates of light were measured with an Apogee PS200 spectroradiometer (Apogee Instruments, Logan, UT) and SpectraWiz spectroscopy software (StellarNet, Tampa, FL). Images of representative seedlings were taken using a Leica MZ FLIII stereo microscope (Leica Microsystems Inc., Buffalo Grove, IL) and processed using Adobe Photoshop CC (Adobe Inc., Mountain View, CA). For hypocotyl measurements, seedlings were scanned with an Epson Perfection V700 photo scanner, and hypocotyl length was measured using the NIH ImageJ software. Box-and-whisker plots of hypocotyl measurements were generated using the Prism 8 software (GraphPad Software, San Diego, CA).

**Plasmid construction and generation of transgenic lines**. The PCR primers used to generate the plasmids containing the wild-type CDS sequences of *PIF*s, *PHYA*, and *PHYB* are listed in Supplementary Table 1. All *PIF3* constructs for the yeast transactivation assays were generated by subcloning either the full-length or a fragment of *PIF3* CDS into the EcoRI and SalI sites of the pBridge vector (TaKaRa

Bio USA, San Jose, CA). The *GST-PIF3* and *GST-PIF3mAD* constructs for GST pulldown assays were generated by subcloning *PIF3* or *PIF3mAD* into the EcoRI and XhoI sites of the pET42b vector (Promega, Madison, WI). The *PIF3-p::HA-YFP-PIF3* and *PIF3p::HA-YFP-PIF3mAD* constructs were generated by subcloning a 2-kb *PIF3* promoter, *3HA-YFP*, and *PIF3* or *PIF3mAD* into the SacI and BamHI sites of the *pJHA212G-RBCSt* vector using HiFi Assembly (New England Biolabs, Ipswich, MA). The vectors for examining the transactivation activities of the PIF3 paralogs were generated by subcloning the full-length CDS sequences of the PIF3 paralogs into the EcoRI and SalI sites of the pBridge vector using HiFi Assembly. The constructs for testing the function of PHYB and PHYA in inhibiting the transactivation activity of PIF3 and PIF1 were generated by subcloning the DNA sequences encoding full-length PHYB or PHYA fused with SV40 NLS, PHYB-C (PHYB amino acids 594–1172 fused with SV40 NLS), and NGB (PHYB amino acids 1–651 fused with GFP, GUS, and SV40 NLS) into the NotI and NdeI sites in the pBridge-PIF3-N4, pBridge-PIF3-M1, pBridge-PIF3, or pBrige-PIF1 plasmid vectors.

The constructs containing mutants of *PIFs* or *PHYB* were generated by one of the following two methods. For PIF3 alanine scanning mutants m1 to m15, the mutations were generated in pBridge-PIF3-N1 with the Q5 Site-Directed Mutagenesis Kit (New England Biolabs, Ipswich, MA). The primers used to generate the m1 to m15 mutants are listed in Supplementary Table 2. For the rest of the *PIF3* and *PHYB* point mutant constructs, two new primers from either the sense or antisense strands harboring the mutations were designed and used in combination with the primers flanking the 5′ and 3′ ends of the respective CDS sequences (Supplementary Table 1) to amplify two overlapping fragments of the respective CDS sequences. The two overlapping DNA fragments were then ligated to the indicated vectors using HiFi Assembly. The pairs of primers containing the mutated nucleotide sequences are listed in Supplementary Table 2.

**Yeast transactivation assay**. Cell viability assays were performed by using either Y2HGold yeast strains (TaKaRa Bio USA, San Jose, CA) containing a pBridge bait vector or diploid yeast strains generated by mating a Y2HGold strain containing a bait vector with a Y187 strain (TaKaRa Bio USA, San Jose, CA) containing the pGADT7 vector. Overnight yeast cultures were diluted to an $OD_{600}$ of 0.2 and serially diluted from $10^0$ to $10^{-4}$. Ten microliters of serial dilutions were spotted onto SD/-Trp media (Y2HGold strains) or SD/-Trp/-Leu (diploid strains) with or without 125 ng/ml AbA. The plates were incubated at 30 °C, and pictures were taken on the third day after plating. Liquid β-galactosidase assays were performed as described in the Yeast Protocols Handbook (TaKaRa Bio USA, San Jose, CA). A Y2HGold yeast strain containing a bait vector was mated with the Y187 strain containing the pGADT7 vector and selected on SD/-Trp/-Leu media. The diploid yeast cells were cultured in liquid SD/-Trp/-Leu media overnight, and the activity of β-galactosidase was measured by using ortho-nitrophenyl-β-galactoside (ONPG) as a substrate. To determine the effect of PHYB on PIF3's transactivation activity, yeast strains containing indicated pBridge vector were grown in SD/-Trp/-Leu/-Met overnight and incubated with 20 μM phycocyanobilin for 4 h in the dark[73]. The yeast cells were then washed with SD-Leu/-Trp/Met media to remove unincorporated phycocyanobilin and incubated in a total of 5 mL of YPDA in 10 μmol m$^{-2}$ s$^{-1}$ of either R or FR light for 6 h. The activity of β-galactosidase was measured by using ONPG as a substrate.

**GST pulldown**. The GST pulldown assay was performed as described previously[74]. GST-PIF3 and GST-PIF3mAD fusion proteins were expressed in *E. coli* strain BL21 (DE3). Cells were harvested via centrifugation and resuspended in E buffer containing 50 mM Tris-HCl pH 7.5, 100 mM NaCl, 1 mM EDTA, 1 mM EGTA, 1% DMSO, 2 mM DTT, and cOmplete™ Protease Inhibitor Cocktail (Sigma-Aldrich, St. Louis, MO). Cells were lysed via French press, and the lysate was centrifuged at $20,000 \times g$ for 20 min at 4 °C. The proteins were precipitated with 3.3 M ammonium sulfate via incubation for 4 h at 4 °C and centrifuged at $10,000 \times g$ for 20 min at 4 °C. The protein pellets were resuspended in E buffer. Insoluble proteins were further cleared by ultracentriguation at $50,000 \times g$ for 15 min at 4 °C. The resulting lysates were dialyzed in E buffer overnight at 4 °C. To immobilize GST fusion proteins, protein lysates were incubated with glutathione Sepharose beads (MilliporeSigma, Burlington, MA) in E buffer for 2 h. Then, the beads were washed with E wash buffer containing 0.1% IGEPAL CA-630. Apoproteins of PHYA-HA and PHYB-HA were prepared using a TNT T7 Coupled Reticulocyte Lysate system (Promega, Madison, WI) as described previously[74]. Holoproteins of PHYA-HA and PHYB-HA were generated by incubating with 20 μM PCB for 1 h in the dark. The in vitro-translated proteins were incubated with the immobilized GST fusion proteins in E wash buffer for 2 h at 4 °C. The beads were washed with E wash buffer, the bound proteins were eluted by boiling in Laemmli sample buffer, and the samples were subjected to SDS-PAGE.

**Protein extraction and immunoblots**. Total proteins from 100 mg of 4-day-old seedlings were extracted in 300 μl of extraction buffer (100 mM Tris-HCl pH 7.5, 100 mM NaCl, 5 mM EDTA pH 8.0, 5% SDS, 20% glycerol, 20 mM DTT, 40 mM β-mercaptoethanol, 2 mM PMSF, cOmplete™ Protease Inhibitor Cocktail, 80 μM MG132 and 80 μM MG115, 1% phosphatase inhibitor cocktail 3, 10 mM

N-ethylmaleimide, and 0.01% bromophenol blue). Samples were immediately boiled for 10 min and centrifuged at $16,000 \times g$ for 10 min. Proteins were separated via SDS-PAGE and blotted onto a nitrocellulose membrane. The membrane was first probed with the indicated primary antibodies and then incubated with goat anti-rabbit (Bio-Rad, cat. no. 1706515) secondary antibodies conjugated with horseradish peroxidase at a 1:5000 dilution. The signals were detected via a chemiluminescence reaction using the SuperSignal West Dura Extended Duration Substrate (ThermoFisher Scientific, Waltham, MA). Rabbit polyclonal anti-PIF3[75], goat polyclonal anti-HA (GenScript, cat. no. A00168), rabbit polyclonal anti-GFP (Abcam, cat. no. ab290), and rabbit polyclonal anti-RPN6 (Enzo Life Sciences, cat. no. BML-PW8370-0100) antibodies were used at a 1:1000 dilution.

**Confocal imaging**. Confocal analysis of HA-YFP-PIF3 and HA-YFP-PIF3mAD was performed as previously described with minor modifications[76]. Seedlings were fixed in 2% paraformaldehyde in PBS under vacuum for 15 min and then was hed three times with 50 mM NH$_4$Cl in PBS for 5 min three times. The seedlings were permeabilized with 0.2% Triton X-100 in PBS for 5 min, incubated with 300 nM DAPI in PBS for 10 min. The seedlings were washed three times with PBS for 5 min three times, and mounted with ProLong™ Diamond Antifade (ThermoFisher Scientific, Waltham, MA) on a slide. The slides were left to cure overnight in the dark, sealed with nail polish, and stored at 4 °C until imaging. Nuclei of hypocotyl epidermal cells were imaged using a Zeiss Axio Observer Z1 inverted microscope equipped with a Plan-Apochromat 100×/1.4 oil-immersion objective and an Axiocam 506 mono camera (Carl Zeiss, Jena, Germany). Fluorescence was detected with the following Zeiss filter sets: YFP, exciter 500/25 nm/nm, emitter 535/40 nm/nm (Zeiss Filter Set 46); DAPI, exciter 365 nm, emitter 445/50 nm/nm (Zeiss Filter Set 49). Images were collected using Zeiss ZEN software and processed using Adobe Photoshop Creative Cloud.

**RNA extraction and qRT-PCR**. Total RNA was extracted from seedlings using a Quick-RNA MiniPrep kit with on-column DNase I treatment (Zymo Research, Irvine, CA). cDNA synthesis was performed with 2 μg of total RNA using oligo(dT) primers and Superscript II First-Strand cDNA Synthesis Kit (ThermoFisher Scientific, Waltham, MA). Quantitative RT-PCR was performed with FastStart Universal SYBR Green Master Mix and a LightCycler 96 Real-Time PCR System (Roche, Basel, Switzerland). Transcript levels of genes were calculated relative to the level of *PP2A*. Genes and primer sets used for qRT-PCR are listed in Supplementary Table 3.

**Chromatin immunoprecipitation**. ChIP assays were performed using chromatin isolated from 4-d-old dark-grown *PIF3/pifq* and *PIF3mAD/pifq* transgenic lines. Seedlings were ground in liquid nitrogen and resuspended in nuclear isolation buffer (10 mM HEPES pH 7.6, 1 M sucrose, 5 mM KCl, 5 mM MgCl$_2$, 5 mM EDTA, 14 mM β-mercaptoethanol, protease inhibitor cocktail, 40 μM MG132, and 40 μM MG115) containing 1% formaldehyde, 0.6% Triton X-100, and 0.4 mM PMSF. Samples were incubated at room temperature for 10 min for cross-linking, and then 125 mM glycine was added to terminate the cross-linking. The lysate was filtered through two-layer Miracloth, and the cleared lysate was centrifuged at $3000 \times g$ for 10 min. The nuclei pellet was resuspended in nuclear isolation buffer, loaded on top of a 15% Percoll solution (15% Percoll, 10 mM HEPES pH 8.0, 1 M sucrose, 5 mM KCl, 5 mM MgCl$_2$, 5 mM EDTA), and centrifuged at $3000 \times g$ for 5 min. The enriched nuclear pellet was lysed with nuclear lysis buffer (50 mM Tris-HCl pH 7.5, 1% SDS, 10 mM EDTA). Then, sonication was performed using a Covaris S2 ultrasonicator (Covaris, Inc., Woburn, MA). The lysate was centrifuged at $13,000 \times g$ for 3 min at 4 °C to remove any debris. The nuclear lysate was diluted with ChIP dilution buffer (15 mM Tris-HCl pH 7.5, 150 m NaCl, 1 mM EDTA, 1% Triton X-100). The lysate was incubated with 1 μg of rabbit polyclonal anti-HA (Abcam, cat. no. ab9110) or rabbit IgG (Cell Signaling Technology, cat. no. 2729S) for 3 h at 4 °C. Immunoprecipitated chromatin with anti-HA antibody or IgG was then incubated with Dynabeads Protein G (ThermoFisher Scientific, Waltham, MA) for 2 h. The beads were washed with ChIP dilution buffer, low-salt wash buffer (20 mM Tris-HCl pH 8.0, 150 mM NaCl, 0.1% SDS, 1% Triton X-100, 2 mM EDTA), high-salt wash buffer (20 mM Tris-HCl pH 8.0, 500 mM NaCl, 0.1% SDS, 1% Triton X-100, 2 mM EDTA), LiCl wash buffer (10 mM Tris-HCl pH 8.0, 0.25 M LiCl, 1% IGEPAL CA-630, 1% sodium deoxycholate, 1 mM EDTA), and TE buffer (10 mM Tris-HCl pH 8.0, 1 mM EDTA). The chromatin was eluted from the beads with elution buffer (1% SDS, 0.1 M NaHCO$_3$). Reverse cross-linking was performed by adding 20 μl of 5 M NaCl to the eluates and incubating at 65 °C overnight. Then, the immuno-precipitated proteins were digested by adding 10 μl of 0.5 M EDTA pH 8.0, 20 μl of 1 M Tris-HCl pH 6.5 and 20 μg Proteinase K and incubating at 50 °C for 2 h. The final chromatin was purified with ChIP DNA Clean & Concentrator (Zymo Research, Irvine, CA). Primer sets used for ChIP-qPCR are listed in Supplementary Table 4.

**Reporting summary**. Further information on research design is available in the Nature Research Reporting Summary linked to this article.

## Data availability

*Arabidopsis* transgenic lines as well as plasmids generated during the current study are available from the corresponding author upon reasonable request. Source data are provided with this paper.

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

## Acknowledgements

We thank Dr. Peter Quail for providing the *pifq* mutant. We thank Dr. Elise Pasoreck for providing valuable comments and suggestions on the manuscript. This work was supported by National Institute of General Medical Sciences Grants R01GM087388 and R01GM132765 to M.C. and National Institute of Food and Agriculture hatch projects CA-R-BPS-5186-H and CA-R-BPS-5084-H to M.C. and X.C., respectively. Q.S. was supported by Guangdong Innovation Team Project 2014ZT05S078.

## Author contributions

C.Y.Y., J.H., Q.S., Y.Q., L.L., B.M., X.C., and M.C. conceived the original research plan; M.C. and X.C. supervised the experiments; C.Y.Y., J.H., Q.S., Y.Q., L.L., R.J.K., E.G.C., and J.H. performed the experiments; C.Y.Y., J.H., Q.S., Y.Q., L.L., R.J.K., E.G.C., J.H., N.M., P.Z., L.C.S., A.N., B.M., X.C., and M.C. analyzed the data; A.N. provided the transgenic lines expressing NGB and NGB mutants; M.C., C.Y.Y., and Q.S. wrote the article with contributions from all authors.

## Competing interests

The authors declare no competing interests.
