## [Peer Review File · Nature Communications]

Direct photoresponsive inhibition of a p53-like transcription activation domain in PIF3 by Arabidopsis phytochrome BREVIEWER COMMENTS

Reviewer #1 (Remarks to the Author):

The paper by Yoo, Sang, He et al. addresses two important and intrinsically related questions. The first of these issues involves the identification of the transcription activation domain (TAD) of the Arabidopsis PIF3 transcription factor. The second issue is whether phytochrome B, which negatively regulates PIFs, interferes with the activity of this TAD. PIFs are transcription factors of major importance in the control of plant growth and development.

The authors use deletion and alanine-scanning mutagenesis in yeast to identify the TAD of PIF3 fused to the Gal4 DNA binding domain (Fig. 1). Then, they compare the amino acid sequence of PIF3 TAD and its surroundings with that of PIF3 proteins in other species and with other PIFs of Arabidopsis (Fig. 2). They introduce mutations of the sites equivalent to the PIF3 TAD in other PIFs to investigate whether they have transactivation activity and whether this is affected by alanine substitutions of the site equivalent to PIF3 TAD (Fig. 2). PIF3 mutated in the five TAD amino acids (PIF3mTAD) retains apparently normal nuclear localization, normal binding to a target gene promoter, but significantly reduced biological activity (Fig. 4-6). PIF3mTAD also retains interaction with active phytochrome B (Fig. 3) and its abundance declines in the light (Fig. 5). When co-expressed in yeast, light-activated phytochrome B abolished the transactivation activity of a fragment of PIF3 containing the TAD and the binding site of phytochrome B but a version of PIF3 lacking this binding site was unaffected. In general, the experiments are well designed and the evidence presented in the paper supports the proposed conclusions. There are, however, some issues that require consideration.

First, in Fig. 4 PIF3mTAD/pifq transgenics appear to have more protein than PIF3/pifq. This is acceptable because more protein in combination with less biological effect supports the conclusion of the authors. However, in Fig. 5 it looks the other way round. These differences are likely random variability between one blot and the other. However, this casts doubts on the previous conclusion. Please provide quantitative data of these protein blots based on three biological replicates as for the rest of the data in the paper. Such information will also help to elucidate whether the rate of decay in the light is similar for PIF3 and PIF3mTAD (Fig. 5b)

Second, statistical tests are not shown in most data displays. Please provide the statistical significance of the differences based on adequate tests. The fold-changes are illustrative but the statistical significance is more important. By looking at the differences and the SD I do not think that the statistical tests will force any major change in the conclusions but they should be included.

Third, under the heading “Relationship between PIF3 transactivation activity and DNA binding” (line 497) the authors discuss the possibility that the previously described effect of phytochrome B on the ability to bind its targets could result from its interference with PIF3 TAD. “... blocking the activity of PIF3’s TAD by PHYB may disrupt the interactions of PIF3 with transcriptional coactivators, thereby reducing PIF3’s association with target gene promoters”. However, the results of fig. 4f appear to contradict this possibility as PIF3mTAD shows normal binding.

My interpretation is that phytochrome B reduces PIF3 activity by three mechanisms. Blockage of PIF3 transactivation activity would be important to repress rapidly PIF3 already bound to DNA. Sequestration would rapidly reduce binding to DNA. Degradation would then reduce PIF3 levels.

Fourth, PIF3mTAD retains substantial PIF activity in planta despite lacking its TAD. Any comment on the possible mechanisms of this residual activity?

Minor issues

Line 44. It is true that the proportion of far-red increases during twilight. However, I do not know any report demonstrating that plants respond to that change.

Lines 91-93. The case of PIF7 does not provide an acceptable argument because changes in the proportion of far-red light modify its nucleo-cytoplasmic partitioning (eLife. 2018; 7: e31636).

Lines 95-97. Suggesting that the control of TAD by phytochrome would be important under special condition where PIF3 is stabilised actually reduces the significance of the mechanism described here. Direct inhibition of PIF3 transcription activation capacity would be a very rapid mode of control. I do not think any other justification is better than that.

Reviewer #2 (Remarks to the Author):

The manuscript submitted by Yoo et al., addresses an essential, much debated and not well understood aspect of the action of the red/far-red light absorbing plant photoreceptors phytochromes. The authors quite convincingly argue in the introduction that phytochromes, more precisely phytochrome-B (phyB), the most important phytochrome species, is likely to govern photomorphogenesis not only by regulating the abundance of phytochrome-interacting-factors (PIF1-8) by initiating their degradation but also by

controlling activity of these bHLH type transcription factors. It has been long ago established that these PIFs interact with phyB in a light dependent fashion and it was the Chen laboratory who has recently showed that in contrast to the assumption the APB motif mediated binding of PIFs to the N-terminal region of photo-activated phyB does not play a major role in light induced degradation of PIFs. To define the biological function of this interaction the authors performed a quite complex set of experiments and concluded that phyB Pfr by binding to the ABP domain of PIF3 directly inhibits the transcription activation domain (TAD) of PIF3. This is a novel, original finding, substantially improves our understanding about the molecular machinery by which light transformed into a signal controlling gene expression.

1./ The authors on their way to pile up evidence to support the above described main conclusion first corroborated that PIFs are potent transcription activators in yeast, then defined the single transactivation domain (TAD) of PIF3 and showed that this exhibits quite significant homology to the TADs of the mammalian p53 and yeast Gcn4 activators. They determined the critical core sequence of PIF3TAD and found that conservation of the critical amino acids of PIF3TAD, except in PIF7, is not particularly high among PIF1-8 although several members of the PIF family were found to be potent activators in yeast. In general, this reviewer finds no major flaw in experiments associated with this part of the manuscript and also accepts the conclusions drawn by the authors.

2./ Next the authors showed that mutation of PIF3TAD compromises functions of PIF3 during skotomorphogenesis. They used the quadruple pif mutant (pif1, pif3, pif4, pif5) to circumvent overlapping functions of PIFs. Fig.4. convincingly shows that mutation of PIF3TAD indeed strongly down regulates expression of select target genes without affecting DNA binding activity of PIF3. It is to be noticed that compared to gene expression effect of mutating PIF3 TAD was very modest on regulating hypocotyl length. The authors hypothesize that PIF3 might control indirectly, via association with other transactivators hypocotyl growth. In this aspect it would be interesting to learn what other transcription activators the authors assume to associate with PIF3 in the given genetic background? Second, to appreciate the "true" contribution of endogenous PIF3 in regulating skotomorphogenesis, the comparison of the endogenous level of PIF3 protein to PIF3-HA-YFP as well comparison of the hypocotyl phenotype of the triple pif1, pif4, pif5 to that of the quadruple pif mutant would be quite helpful.

3./ The authors may wish to take into consideration the very same comments regarding the experiments illustrating the effect of mutating PIF3TAD in regulating expression of ELIP2

4./ The presented data documenting the compromising effect of mutating PIF3TAD for PIF3 function in gene activation by shade are satisfactory yet the reviewer still has concerns regarding the true, in vivo contribution of the endogenous PIF3 to this particular response (see above). PIF7 had been documented to play a significant role in regulating shade responses and the authors found that PIF7TAD shows the highest homology to PIF3TAD. Was there any particular reason to omit manipulation of PIF7TAD for defining the effect of photoinhibition of PIF7 transactivation activity in regulating shade responses?

5./Figure 7 contains the data of the most critical experiments regarding the main conclusion of the ms. The authors made use of the mutant transgenic lines generated by the Nagatani laboratory some years ago. The data shown are compelling yet this reviewer would like to make some remarks. The original paper by Oka et.al also describes a fourth NGB mutant, termed G112D whose phenotype was aborted when the mutation was introduced into the full length phyB. Did the authors investigate this NGB mutant and if so what the outcome? The Nagatani laboratory produced full length phyBs harboring the used mutations. It would be interesting and informative to see the endogenous PIF3 and phyB levels in the lines that contain the full length phyB variants. This may help to make some educated guesses regarding the impact of PIF3 degradation (co-degradation with phyB?) and photoinhibition of PIF3TAD activity. Figure 7 is quite busy and this reviewer missed to find the data showing hypocotyl length to which the figure legend refers.

Beside the above comments this reviewer would like to make some additional remarks. Personally I am convinced that beside degradation of PIF3 protein, photoinhibition of PIF3 transactivation activity is indeed contributing to the regulation of PIF3 biological function in skoto and/or photomorphogenesis. However, the vexing question is whether DNA bound PIF3 can be phosphorylated and/or does phosphorylation of PIF3 prevent its binding to DNA and last but not least are these events mutually exclusive or not. In addition, I also assume that dynamics of degradation of the PIF3 protein and photoinhibition of PIF3 transactivation activity is likely to play an important role in regulating particular photoresponses. Phosphorylation sites of PIF3 had been studied in detail by the Quail laboratory (in light and dark) (Ni.W et.al., Plant Cell 2013). Did the authors consider to test the effect of selected PIF3 phospho-mutants on the PIF3 transactivation activity, DNA binding and if not why? Finally, this reviewer notes that the authors showed that mutation of PIF3 TAD does not affect interaction of PIF3 either with PHYB or PHYA. PIF1 TAD displays a reasonably high homology to the defined PIF3TAD. PIF1 is an important player in phyA initiated signaling. The authors have the required mutants and tools to define whether photoinhibition of PIF3 and/or PIF1 transactivation activity by phyA occurs. Including even just a limited set of such experiments would further increase the impact of the paper.

Response to Reviewers

We thank the reviewers for their positive comments and valuable suggestions.

Reviewer #1

The paper by Yoo, Sang, He et al. addresses two important and intrinsically related questions. The first of these issues involves the identification of the transcription activation domain (TAD) of the Arabidopsis PIF3 transcription factor. The second issue is whether phytochrome B, which negatively regulates PIFs, interferes with the activity of this TAD. PIFs are transcription factors of major importance in the control of plant growth and development.

The authors use deletion and alanine-scanning mutagenesis in yeast to identify the TAD of PIF3 fused to the Gal4 DNA binding domain (Fig. 1). Then, they compare the amino acid sequence of PIF3 TAD and its surroundings with that of PIF3 proteins in other species and with other PIFs of Arabidopsis (Fig. 2). They introduce mutations of the sites equivalent to the PIF3 TAD in other PIFs to investigate whether they have transactivation activity and whether this is affected by alanine substitutions of the site equivalent to PIF3 TAD (Fig. 2). PIF3 mutated in the five TAD amino acids (PIF3mTAD) retains apparently normal nuclear localization, normal binding to a target gene promoter, but significantly reduced biological activity (Fig. 4-6). PIF3mTAD also retains interaction with active phytochrome B (Fig. 3) and its abundance declines in the light (Fig. 5). When co-expressed in yeast, light-activated phytochrome B abolished the transactivation activity of a fragment of PIF3 containing the TAD and the binding site of phytochrome B but a version of PIF3 lacking this binding site was unaffected. In general, the experiments are well designed and the evidence presented in the paper supports the proposed conclusions. There are, however, some issues that require consideration.

First, in Fig. 4 PIF3mTAD/pifq transgenics appear to have more protein than PIF3/pifq. This is acceptable because more protein in combination with less biological effect supports the conclusion of the authors. However, in Fig. 5 it looks the other way round. These differences are likely random variability between one blot and the other. However, this casts doubts on the previous conclusion. Please provide quantitative data of these protein blots based on three biological replicates as for the rest of the data in the paper. Such information will also help to elucidate whether the rate of decay in the light is similar for PIF3 and PIF3mTAD (Fig. 5b)

Response: We thank the reviewer for the comments. We have quantified the protein levels in the immunoblots in Fig. 4a, 5b, and 6c. The data consistently show that the two *PIF3mTAD/pifq* lines had slightly less HA-YFP-PIF3mTAD than the levels of HA-YFP-PIF3 in the two *PIF3/pifq* lines. However, these small variations cannot account for the significant differences in the expression of PIF3 target genes between the *PIF3/pifq* and *PIF3mTAD/pifq* lines. Also, we added quantification data from four independent experiments to show HA-YFP-PIF3mTAD and HA-YFP-PIF3 had the same degradation rate during the dark-to-light transition (Fig. 5b).

Second, statistical tests are not shown in most data displays. Please provide the statistical significance of the differences based on adequate tests. The fold-changes are illustrative but the statistical significance is more important. By looking at the differences and the SD I do not think that the statistical tests will force any major change in the conclusions but they should be included.

Response: We added statistical tests to the liquid β -galactosidase assays in Fig. 1a, 1b, 2c, the ChIP results in Fig. 4f. The rest of the data had included statistical tests in the previous version.

Third, under the heading “Relationship between PIF3 transactivation activity and DNA binding” (line 497) the authors discuss the possibility that the previously described effect of phytochrome B on the ability to bind its targets could result from its interference with PIF3 TAD. “... blocking the activity of PIF3’s TAD by PHYB may disrupt the interactions of PIF3 with transcriptional coactivators, thereby reducing PIF3’s association with target gene promoters”. However, the results of fig. 4f appear to contradict this possibility as PIF3mTAD shows normal binding. My interpretation is that phytochrome B reduces PIF3 activity by three mechanisms. Blockage of PIF3 transactivation activity would be important to repress rapidly PIF3 already bound to DNA. Sequestration would rapidly reduce binding to DNA. Degradation would then reduce PIF3 levels.

Response: The reviewer raised an important point. The previous studies by Dr. Giltsu Choi’s group showed that PIF3’s DNA binding is attenuated by photoactivated PHYB. However, the experiments in Fig. 4f was performed in the true dark, where PHYB was in the inactive form and localized to the cytoplasm. This experiment was designed to show whether the mTAD mutations could have an effect on PIF3 DNA binding (in the absence of any influence by active phytochromes). We agree with the reviewer’s view. We discussed these possibilities in Discussion. Sequestration of PIF3 away from target-gene promoters could be one of the mechanisms.

Fourth, PIF3mTAD retains substantial PIF activity in planta despite lacking its TAD. Any comment on the possible mechanisms of this residual activity?

Response: There could be at least a couple of possibilities attributing to the remaining activity of PIF3mTAD. First, the PIF3mTAD mutant may retain some transactivation activity *in planta*. Second, it is possible that the transactivation activity of PIF3 for some genes could be contributed by not only PIF3 but also PIF3-associated transcription factors. Although PIF3mTAD loses the transactivation activity, it can still bind DNA and likely help recruit other transcription factors. Consistent with the latter possibility, genome-wide studies have shown that the PIF-binding sites are frequently associated with the binding sites of many other transcription factors, some of which have been shown to directly interact with PIFs [Kim et al. (2016) *Plant Cell* 28:1388-1405; Zhang et al. (2019) *PNAS* 117:3261-69]. Reviewer #2 also asked the same question. We discussed these possibilities in the revised manuscript.

Minor issues

Line 44. It is true that the proportion of far-red increases during twilight. However, I do not know any report demonstrating that plants respond to that change.

Response: The timing of far-red increases during sunrise and sunset in combination with temperature are critical indicators of seasonal changes. We really like the following three

references: Franklin and Whitelam (2007) *Nat Genet* 39:1410-1412, Salter MG et al. (2003) *Nature* 426:680-683, and Song YH et al. (2018) *Nat Plants* 4:824-835.

Lines 91-93. The case of PIF7 does not provide an acceptable argument because changes in the proportion of far-red light modify its nucleo-cytoplasmic partitioning (eLife. 2018; 7: e31636).

Response: A recent paper from Joanne Chory lab showed that PIF7 is sequestered by PHYB on photobodies to attenuate the transactivation activity of PIF7 [Willege BC et al. (2021) *Nat Genet* 53:955-961]. Therefore, it is possible that the transactivation activity of PIF7 is repressed by PHYB on photobodies.

Lines 95-97. Suggesting that the control of TAD by phytochrome would be important under special condition where PIF3 is stabilised actually reduces the significance of the mechanism described here. Direct inhibition of PIF3 transcription activation capacity would be a very rapid mode of control. I do not think any other justification is better than that.

Response: We really appreciate the comment and have revised this part to the following: “Compared with the mechanism of PIF degradation, direct inhibition of PIFs’ transactivation activity provides a rapid mode of photoresponsive gene regulation, and also, it is likely a critical mechanism to regulate the PIFs that can accumulate in the light, such as PIF4, PIF5, and PIF³⁰⁻³³. ”

Reviewer #2:

The manuscript submitted by Yoo et al., addresses an essential, much debated and not well understood aspect of the action of the red/far-red light absorbing plant photoreceptors phytochromes. The authors quite convincingly argue in the introduction that phytochromes, more precisely phytochrome-B (phyB), the most important phytochrome species, is likely to govern photomorphogenesis not only by regulating the abundance of phytochrome-interacting-factors (PIF1-8) by initiating their degradation but also by controlling activity of these bHLH type transcription factors. It has been long ago established that these PIFs interact with phyB in a light dependent fashion and it was the Chen laboratory who has recently showed that in contrasts to the assumption the APB motif mediated binding of PIFs to the N-terminal region of photo-activated phyB does not play a major role in light induced degradation of PIFs. To define the biological function of this interaction the authors performed a quite complex set of experiments and concluded that phyB Pfr by binding to the ABP domain of PIF3 directly inhibits the transcription activation domain (TAD) of PIF3. This is a novel, original finding, substantially improves our understanding about the molecular machinery by which light transformed into a signal controlling gene expression.

1. The authors on their way to pile up evidence to support the above described main conclusion first corroborated that PIFs are potent transcription activators in yeast, then defined the single transactivation domain (TAD) of PIF3 and showed that this exhibits quite significant homology to the TADs of the mammalian p53 and yeast Gcn4 activators. They determined the critical core sequence of PIF3TAD and found that conservation of the critical amino acids of PIF3TAD, except in PIF7, is not particularly high among PIF1-8 although several members of the PIF family were found to be potent activators in yeast. In general, this reviewer finds no major flaw in experiments associated with this part of the manuscript and also accepts the conclusions drawn by the authors.

Response: we thank the reviewer for the positive assessment of the manuscript.

2. Next the authors showed that mutation of PIF3TAD comprises functions of PIF3 during skotomorphogenesis. They used the quadruple *pif* mutant (*pif1*, *pif3*, *pif4*, *pif5*) to circumvent overlapping functions of PIFs. Fig.4. convincingly shows that mutation of PIF3TAD indeed strongly down regulates expression of select target genes without affecting DNA binding activity of PIF3. It is to be noticed that compared to gene expression effect of mutating PIF3 TAD was very modest on regulating hypocotyl length. The authors hypothesize that PIF3 might control indirectly, via association with other transactivators hypocotyl growth. In this aspect it would be interesting to learn what other transcription activators the authors assume to associate with PIF3 in the given genetic background? Second, to appreciate the “true” contribution of endogenous PIF3 in regulating skotomorphogenesis, the comparison of the endogenous level of PIF3 protein to PIF3-HA-YFP as well comparison of the hypocotyl phenotype of the triple *pif1*, *pi4*, *pif5* to that of the quadruple *pifq* mutant would be quite helpful. The authors may wish to take into consideration the very same comments regarding the experiments illustrating the effect of mutating PIF3TAD in regulating expression of *ELIP2*.

Response: There could be at least a couple of possibilities attributing to the remaining activity of PIF3mTAD. First, it is possible that the PIF3mTAD mutant retains some transactivation activity *in planta*. Second, the transactivation activity of PIF3 for some genes could be contributed by not only PIF3’s TAD but also the TADs of other transcription factors associated with PIF3. Although PIF3mTAD loses its TAD activity, it can still bind DNA and likely retains its activity to interact with other transcription factors, therefore, it could still recruit other activators to activate gene expression to a certain degree. Consistent with the latter possibility, it has been shown that the PIF-binding sites genome-wide are frequently associated with the binding sites of many other transcription factors, some of which have been shown to directly interact with PIFs [Kim et al. (2016) *Plant Cell* 28:1388-1405; Zhang et al. (2020) *PNAS* 117:3261-69].

As suggested by the reviewer, we added the *pif145* triple mutant in Fig. 4. The results show that, although the levels of HA-YFP-PIF3 were significantly higher than the level of endogenous PIF3 in *pif145*, the hypocotyl phenotypes of the *PIF3/pifq* lines were similar to that of *pif145* in darkness, supporting the conclusion that the *PIF3/pifq* lines rescued the function of the endogenous PIF3. With that said, the expression of the PIF3 target genes, *PIL1*, *XTR7* and *RD20*, was considerably higher in the *PIF3/pifq* transgenic lines compared with *pif145* likely due to the higher levels of HA-YFP-PIF3. However, in clear contrast, the *PIF3mTAD/pifq* lines, which had similar levels of HA-YFP-PIF3mTAD compared with the levels of HA-YFP-PIF3 in the *PIF3/pifq* lines, showed lower transcript levels of the PIF3 target genes compared with those in *pif145*. Together, these results are consistent with the conclusion that PIF3mTAD is impaired in the transactivation activity of PIF3. We did not perform additional experiments for *ELIP2* activation in the *pif145* mutant, because unlike the weak hypocotyl phenotype of the *pif3* mutant, induction of *ELIP2* has been well defined as a PIF3-dependent process in the *pif3* single mutant [Al-Sady et al. (2008) *PNAS* 105:2232-2237].

3. The presented data documenting the compromising effect of mutating PIF3TAD for PIF3 function in gene activation by shade are satisfactory yet the reviewer still has concerns regarding the true, *in vivo* contribution of the endogenous PIF3 to this particular response (see above). PIF7 had been documented to play a significant role in regulating shade responses and the authors found that PIF7TAD shows the highest homology to PIF3TAD. Was there any particular reason to omit manipulation of PIF7TAD for defining the effect of photoinhibition of PIF7 transactivation activity in regulating shade responses?

Response: We thank the reviewer for the comment. We agree that PIF4, 5, and 7 also play major roles in the shade response. The transgenic lines with much higher levels of recombinant PIF3 proteins do not fully reflect the contributions of PIF3 in the process, we have added the following in the paragraph: “It is important to note that our experimental design was not aimed at assessing the contribution of PIF3 in shade responses; because PIF3 accumulates to higher levels in our transgenic lines, the contribution of PIF3 to shade responses in our experimental settings was likely overestimated. However, our results clearly show that PIF3mTAD is impaired in the transactivation activity under shade treatment.” As explained in the manuscript, because the recombinant PIF3 proteins accumulated in the transgenic lines, it gave us an opportunity to assess the regulation of PIF3 activity by PHYB.

We agree with the reviewer that it would be interesting to evaluate the activity of other PIFs, such as PIF7, by generating transgenic lines expressing their respective mTAD mutants. However, at the time when these experiments were designed, we did not know that the mTAD mutant of PIF3 would behave as expected *in planta*, the main purpose was still to validate the *in vivo* activity of the PIF3 TAD first. So, we did not prepare the mTAD mutants of other PIFs at the same time. Now, with the current data, we agree that it would be very interesting to examine the mTAD mutant of other PIFs, particularly PIF7, in future investigations.

4. Figure 7 contains the data of the most critical experiments regarding the main conclusion of the ms. The authors made use of the mutant transgenic lines generated by the Nagatani laboratory some years ago. The data shown are compelling yet this reviewer would like to make some remarks. The original paper by Oka et.al also describes a fourth NGB mutant, termed G112D whose phenotype was aborted when the mutation was introduced into the full length phyB. Did the authors investigate this NGB mutant and if so what the outcome? The Nagatani laboratory produced full length phyBs harboring the used mutations. It would be interesting and informative to see the endogenous PIF3 and phyB levels in the lines that contain the full length phyB variants. This may help to make some educated guesses regarding the impact of PIF3 degradation (co-degradation with phyB?) and photoinhibition of PIF3TAD activity. Figure 7 is quite busy and this reviewer missed to find the data showing hypocotyl length to which the figure legend refers.

Response: We did not use the G112D mutation because it is the weakest allele among the four mutations, i.e., the G112D mutation did not abolish the PHYB-APB interaction as much as the other three mutations used in the study [Kikis et al. (2009) *PloS Genet* 5:e1000352].

We have characterized the lines (from Akira Nagatani lab) expressing full-length phyB with the mutations abolishing the PHYB-APB interaction. Those data were included in our previous

publication [Qiu et al. (2017) *Nat Commun* 8:1905] to show that the PHYB-APB interaction does not play a major role in PIF3 degradation. Because PIF3 does not accumulate in those lines, they cannot be used to assess the photoinhibition activity of PHYB on the PIF3 TAD.

We agree with the reviewer regarding Figure 7. In the revised manuscript, we split Figure 7 into two figures.

*5. Beside the above comments this reviewer would like to make some additional remarks. Personally I am convinced that beside degradation of PIF3 protein, photoinhibition of PIF3 transactivation activity is indeed contributing to the regulation of PIF3 biological function in skoto and/or photomorphogenesis. However, the vexing question is whether DNA bound PIF3 can be phosphorylated and/or does phosphorylation of PIF3 prevent its binding to DNA and last but not least are these events mutually exclusive or not. In addition, I also assume that dynamics of degradation of the PIF3 protein and photoinhibition of PIF3 transactivation activity is likely to play an important role in regulating particular photoresponses. Phosphorylation sites of PIF3 had been studied in detail by the Quail laboratory (in light and dark) (Ni.W et.al., *Plant Cell* 2013). Did the authors consider to test the effect of selected PIF3 phospho-mutants on the PIF3 transactivation activity, DNA binding and if not why?*

Response: We thank the reviewer for the comment. It is a great idea that the TAD activity of PIF3 could be influenced by phosphorylation. We have not had the opportunity to test the hypothesis. We added the following paragraph in the Discussion: “PHYs trigger rapid phosphorylation of PIF3 by Photoregulatory Protein Kinases (PPKs) in the light^{28,57}. The PIF3 TAD contains two light-dependent phosphorylation sites, S102 and S108⁶³. Phosphorylation at these sites is expected to increase negative charges around the activator motif, which could potentially alter the accessibility of the activator motif and/or the affinity of the TAD to transcriptional coactivators. Therefore, PHYB and PHYA may also modulate the activity of the PIF3 TAD through phosphorylation. This hypothesis needs to be further tested in future studies.”

6. Finally, this reviewer notes that the authors showed that mutation of PIF3 TAD does not affect interaction of PIF3 either with PHYB or PHYA. PIF1 TAD displays a reasonably high homology to the defined PIF3TAD. PIF1 is an important player in phyA initiated signaling. The authors have the required mutants and tools to define whether photoinhibition of PIF3 and/or PIF1 transactivation activity by phyA occurs. Including even just a limited set of such experiments would further increase the impact of the paper.

Response: This is a great idea. We have performed additional experiments to show that both PHYB and NGB inhibit the activity of PIF1 in yeast (Fig. 7d), supporting the idea that PHYB can directly inhibit the activity of PIF3 paralogs. Also, we have added PHYA data to show that PHYA can inhibit the activity of PIF3 and PIF1 in a light-dependent manner as well (Fig. 7e).

REVIEWERS' COMMENTS

Reviewer #1 (Remarks to the Author):

The work by Yoo et al. investigates the mechanisms involved in the control of transcription by phyB and PIF3, a module of fundamental importance in plant responses to the light environment. A Google Scholar search for the last five years reveals nearly 4000 Arabidopsis papers for phyB, more than 2000 for PIF3 and more than 1000 for phyB and PIF3. The authors present a comprehensive set of experiments and the results provide significant novel insight. They convincingly identify the transcription activation domain of PIF3 and show that phyB negatively regulates PIF3 activity by interfering with this domain. Despite its intrinsic complexity, the work is relatively easy to read thanks to the well-organised text and figures. Three very specific issues would require additional consideration. Addressing these issues satisfactorily should involve no major complications.

First, in my previous evaluation, I had noted that PIF3mTAD/pifq transgenics appeared to have more protein than PIF3/pifq. After quantifying the protein blots, the authors concluded in their letter that in fact these differences occur consistently. There, they also explained that these differences in PIF3 were unlikely to justify the differences in gene expression observed among the lines. I think that the explanation is reasonable. However, in my view, the text should briefly mention this issue to make it clear for the reader. Along the same line, the legend to Fig. 5b should indicate that protein levels are relative to the abundance of each line in darkness, but in darkness, HA-YFP-PIF3mTAD had less protein than HA-YFP-PIF3.

Second, I had also mentioned in my previous evaluation that, to my knowledge, there were no reports demonstrating that plants respond to the reduction in red / far-red ratio of the light during sunrise and sunset. In their response, the authors listed three references in support of their statement. They used one of these references in the text. The problem is that none of these three papers supports the conclusion that plants respond to the reduction in red / far-red ratio of the light during sunrise and sunset. The papers by Franklin and Whitelam (2007, Nat Genet 39:1410-1412) and Salter et al. (2003, Nature 426:680-683) investigate the effects of simulated shade and how they depend on the circadian clock. These papers use free-running conditions of continuous light and not the light-dark cycles required to define actual "sunrise" and "sunset". Many genes show very different phases of expression in plants grown under free-running conditions compared to light / dark cycles (see http://diurnal.mocklerlab.org/diurnal_data_finders/new). Subjective sunrise or sunset is not the same as actual sunrise or sunset because the later include changes in the light input, not present under continuous light. Therefore, we cannot use free running conditions to define with precision when a given response occurs under natural conditions. Furthermore, these two papers use very low red / far-red ratios (0.1) because they deal with the responses to shade, not with the response to twilight. The decrease in red / far-red ratio caused by atmospheric factors during sunrise and sunset never reaches such low values. Finally, the paper by Song et al. (2018, Nat Plants 4:824-835) reports the effects of

adding far-red light throughout the photoperiod, not during sunrise or sunset. They actually compare artificial light with a red / far-red ratio of 4 with light of a red / far-red ratio of 1. This means that the Song et al. paper does not even include a red / far-red ratio typical of sunrise or sunset (it is not its aim). I am sorry for this long argument about an issue that is certainly lateral here. However, I think the paper by Yoo et al. is going to capture the attention of a wide audience, which could propagate a concept not sustained by the literature.

Third, on lines 425-426 the authors state that the light-independent interaction between the C-terminal module of phyB and PIF3 promotes PIF3 degradation. How do they conciliate this statement with the fact that the control of PIF3 degradation by phyB is light dependent? During de-etiolation the light-induced migration of phyB to the nucleus to interact with PIF3 could account for the light-dependency of degradation. However, PIF3 also accumulates in response to low red / far-red ratios (Leivar et. Al, 2012, *Molecular Plant* 5, 208-223) How could this happen if phyB remains bound to PIF3 causing its degradation. Would interaction with a third protein provide the light dependency to stability? Perhaps the authors could briefly clarify this issue.

Reviewer #2 (Remarks to the Author):

The authors performed the suggested/requested additional experiments, changed and refined some of the conclusions in harmony with the reviewers requests. These changes further improved the quality of the ms. This reviewer has no additional critical comments that needs to be addressed.

Response to Reviewers

We thank the reviewers for their insightful comments and suggestions.

Reviewer #1

The work by Yoo et al. investigates the mechanisms involved in the control of transcription by phyB and PIF3, a module of fundamental importance in plant responses to the light environment. A Google Scholar search for the last five years reveals nearly 4000 Arabidopsis papers for phyB, more than 2000 for PIF3 and more than 1000 for phyB and PIF3. The authors present a comprehensive set of experiments and the results provide significant novel insight. They convincingly identify the transcription activation domain of PIF3 and show that phyB negatively regulates PIF3 activity by interfering with this domain. Despite its intrinsic complexity, the work is relatively easy to read thanks to the well-organised text and figures. Three very specific issues would require additional consideration. Addressing these issues satisfactorily should involve no major complications.

Response: We thank the reviewer for his/her positive assessment of our work.

First, in my previous evaluation, I had noted that PIF3mTAD/pifq transgenics appeared to have more protein than PIF3/pifq. After quantifying the protein blots, the authors concluded in their letter that in fact these differences occur consistently. There, they also explained that these differences in PIF3 were unlikely to justify the differences in gene expression observed among the lines. I think that the explanation is reasonable. However, in my view, the text should briefly mention this issue to make it clear for the reader. Along the same line, the legend to Fig. 5b should indicate that protein levels are relative to the abundance of each line in darkness, but in darkness, HA-YFP-PIF3mTAD had less protein than HA-YFP-PIF3.

Response: We thank the reviewer for his/her careful assessment.

We added the following sentence in the text: “HA-YFP-PIF3 and HA-YFP-PIF3mAD were expressed at similar levels in the transgenic lines, despite slightly less abundance of HA-YFP-PIF3mAD in the *PIF3mAD/pifq* lines (Fig. 4a).”

We also added the following to the legend of Fig. 5b: “The numbers beneath the PIF3 immunoblot indicate the relative levels of HA-YFP-tagged PIF3 or PIF3mAD relative to the abundance of each line in the darkness. The right panel shows the kinetic changes of HA-YFP-PIF3 and HA-YFP-PIF3mAD during the dark-to-light transition using data from all four experiments. The protein levels of PIF3 and PIF3mAD were quantified relative to those at time 0. Although *PIF3mAD/pifq* accumulated slightly less HA-YFP-PIF3mAD and HA-YFP-PIF3 in *PIF3/pifq* in darkness, the degradation kinetics of HA-YFP-PIF3mAD was same as that of HA-YFP-PIF3.”

Second, I had also mentioned in my previous evaluation that, to my knowledge, there were no reports demonstrating that plants respond to the reduction in red / far-red ratio of the light during sunrise and sunset. In their response, the authors listed three references in support of their statement. They used one of these references in the text. The problem is that none of these three papers supports the conclusion that plants respond to the reduction in red / far-red ratio of the light during sunrise and sunset. The papers by Franklin and Whitelam (2007, Nat Genet 39:1410-1412) and Salter et al. (2003, Nature 426:680-683) investigate the effects of simulated shade and how they depend on the circadian clock. These papers use free-running conditions of continuous light and not the light-dark cycles required to define actual “sunrise” and “sunset”. Many genes show very different phases of expression in plants grown under free-running conditions compared to light / dark cycles (see http://diurnal.mocklerlab.org/diurnal_data_finders/new). Subjective sunrise or sunset is not the same

as actual sunrise or sunset because the later include changes in the light input, not present under continuous light. Therefore, we cannot use free running conditions to define with precision when a given response occurs under natural conditions. Furthermore, these two papers use very low red / far-red ratios (0.1) because they deal with the responses to shade, not with the response to twilight. The decrease in red / far-red ratio caused by atmospheric factors during sunrise and sunset never reaches such low values. Finally, the paper by Song et al. (2018, Nat Plants 4:824-835) reports the effects of adding far-red light throughout the photoperiod, not during sunrise or sunset. They actually compare artificial light with a red / far-red ratio of 4 with light of a red / far-red ratio of 1. This means that the Song et al. paper does not even include a red / far-red ratio typical of sunrise or sunset (it is not its aim). I am sorry for this long argument about an issue that is certainly lateral here. However, I think the paper by Yoo et al. is going to capture the attention of a wide audience, which could propagate a concept not sustained by the literature.

Response: We really appreciate the reviewer for the insightful comments. We have revised the sentence in the Introduction to:

“In particular, alterations in the abundance and ratio of red (R, 600 nm to 700 nm) and far-red (FR, 700 nm to 750 nm) light are powerful environmental cues that inform about space and time – such as the availability of photosynthetically active R-light radiation, the sense of morning and evening as well as seasonal fluctuations in day length, and the threat by neighboring vegetation or shade that depletes R light and alters the R-to-FR ratio.”

Third, on lines 425-426 the authors state that the light-independent interaction between the C-terminal module of phyB and PIF3 promotes PIF3 degradation. How do they conciliate this statement with the fact that the control of PIF3 degradation by phyB is light dependent? During de-etiolation the light-induced migration of phyB to the nucleus to interact with PIF3 could account for the light-dependency of degradation. However, PIF3 also accumulates in response to low red / far-red ratios (Leivar et. Al, 2012, Molecular Plant 5, 208-223) How could this happen if phyB remains bound to PIF3 causing its degradation. Would interaction with a third protein provide the light dependency to stability? Perhaps the authors could briefly clarify this issue.

Response: This is a great question. One possibility is that PIF3 degradation requires the colocalization of PIF3 and phyB on photobodies, where factors required for PIF3 degradation, such as the PIF3 E3 ubiquitin ligases, the LRBs, and the kinases for PIF3 phosphorylation, the PPKs, are concentrated. Although photobody localization of phyB is mainly mediated by the C-terminal output module, the photobody formation activity of the C-terminal output module of phyB can be regulated by the N-terminal photosensory module, which could provide the light-dependency to PIF3 stability. Consistent with this model, NGB (which contains the photosensory module alone) does not localize to photobodies and cannot mediate PIF3 degradation.

Reviewer #2:

The authors performed the suggested/requested additional experiments, changed and refined some of the conclusions in harmony with the reviewers requests. These changes further improved the quality of the ms. This reviewer has no additional critical comments that needs to be addressed.

Response: We thank this reviewer for his/her insightful and valuable suggestions that inspired us to further improve our manuscript.